# Mechanisms of nucleotide selection by telomerase

**Matthew A Schaich[1], Samantha L Sanford[2], Griffin A Welfer[1], Samuel A Johnson[2], Thu H Khoang[1], Patricia L Opresko[2], Bret D Freudenthal[1,3]\***

[1]Department of Biochemistry and Molecular Biology, University of Kansas Medical Center, Kansas City, United States; [2]Department of Environmental and Occupational Health, University of Pittsburgh Graduate School of Public Health, and UPMC Hillman Cancer Center, Pittsburgh, United States; [3]Department of Cancer Biology, University of Kansas Medical Center, Kansas City, United States

**Abstract** Telomerase extends telomere sequences at chromosomal ends to protect genomic DNA. During this process it must select the correct nucleotide from a pool of nucleotides with various sugars and base pairing properties, which is critically important for the proper capping of telomeric sequences by shelterin. Unfortunately, how telomerase selects correct nucleotides is unknown. Here, we determined structures of *Tribolium castaneum* telomerase reverse transcriptase (TERT) throughout its catalytic cycle and mapped the active site residues responsible for nucleoside selection, metal coordination, triphosphate binding, and RNA template stabilization. We found that TERT inserts a mismatch or ribonucleotide ~1 in 10,000 and ~1 in 14,000 insertion events, respectively. At biological ribonucleotide concentrations, these rates translate to ~40 ribonucleotides inserted per 10 kilobases. Human telomerase assays determined a conserved tyrosine steric gate regulates ribonucleotide insertion into telomeres. Cumulatively, our work provides insight into how telomerase selects the proper nucleotide to maintain telomere integrity.

**\*For correspondence:**
bfreudenthal@kumc.edu

**Competing interests:** The authors declare that no competing interests exist.

## Introduction

During every round of eukaryotic cell division, a small amount of DNA is lost from the ends of each chromosome (*Olovnikov, 1973*; *Watson, 1972*). Termed the end replication problem, this phenomenon is countered by two complementary adaptations. First, repetitive noncoding DNA sequences, known as telomeres, are found at chromosomal ends, preventing the loss of vital genetic information during each cell division (*Blackburn and Gall, 1978*; *Moyzis et al., 1988*). Second, the ribonucleoprotein telomerase elongates shortened telomeres at chromosomal ends using a reverse transcriptase activity (*Greider and Blackburn, 1987*). Without elongation, telomeres will eventually reach a critically short length, causing cells to undergo apoptosis or become senescent (*Hayflick and Moorhead, 1961*; *Meyerson, 1998*). Because telomerase plays such a fundamental role in the temporal regulation of cell division, aberrations in telomerase are implicated in numerous human diseases. These include premature aging, idiopathic pulmonary fibrosis (IPF), dyskeratosis congenita, and cancer (*Blasco, 2005*; *Kim et al., 1994*; *Nelson and Bertuch, 2012*). In particular,~90% of cancers upregulate telomerase to combat telomere shortening and enable unlimited cell division, as opposed to somatic cells where telomerase is absent (*Jafri et al., 2016*).

The implication of telomerase in multiple human diseases underscores the importance of understanding its catalytic cycle at the molecular level. Although the human telomerase holoenzyme is composed of multiple accessory subunits, catalysis is localized to the telomerase reverse transcriptase (TERT) subunit (*Nguyen et al., 2018*). Historically, biochemical characterization of human TERT (*h*TERT) has proven challenging, in part because of technical difficulties purifying and reconstituting large quantities of active telomerase (*Ramakrishnan et al., 1997*; *Schmidt et al., 2016*). As a result,

many fundamental parameters describing the telomerase catalytic cycle have remained undefined. These enzymatic constants are essential for understanding the roles of active site residues, the selection of right from the wrong nucleotides (fidelity), and the prevention of ribonucleotide triphosphate (rNTP) insertion (sugar discrimination). The faithful extension of telomeres by telomerase is critical because aberrations in telomeric sequences prevent shelterin proteins from capping telomeres, thus promoting genomic instability (*de Lange, 2005*; *Nandakumar et al., 2010*). Structural studies of human telomerase have also been historically challenging. Currently, the highest resolution structural snapshot of human telomerase is a cryo-EM structure at 8 Å resolution (*Nguyen et al., 2018*). This structure represents a milestone in telomerase structural biology, revealing details of the telomerase tertiary and secondary structure. However, the positions of amino acids are difficult to distinguish at this resolution, leaving many molecular details of the catalytic cycle ambiguous.

To mitigate the difficulties inherent in the biochemical characterization of human telomerase, several model systems have been established. These include models from yeast, the protazoa *Tetrahymena thermophila* (with which a 5 Å cryo-EM structure was recently determined), and the insect model *Tribolium castaneum* (sequence alignment shown in *Figure 1—figure supplement 1*; *Gillis et al., 2008*; *Jiang et al., 2018*; *Petrova et al., 2018*). For biochemical characterization of TERT, we opted to use *T. castaneum* TERT (*tc*TERT) for the following reasons: first, *tc*TERT readily fit into the cryo-EM density of *h*TERT and aligns well with the recent cryo-EM structure from *T. thermophila*, highlighting the conserved secondary structure (*Figure 1—figure supplement 2*); second, upon alignment with *h*TERT, the active site pocket of *tc*TERT exhibits a high degree of sequence identity (*Supplementary file 1*, Table 1a); third, using a truncated version of the *T. castaneum* telomerase RNA component (TR), we can readily obtain sufficient quantities of isolated, active *tc*TERT for characterization of the telomerase catalytic cycle by pre-steady-state kinetics and X-ray crystallography (*Gillis et al., 2008*; *Nguyen et al., 2018*). Although TERTs have highly conserved active sites, there are significant changes in the domain architecture between human and *tc*TERT. These include *tc*TERT lacking the N-terminal (TEN) domain and missing a portion of the insertion in fingers domain (IFD) (*Supplementary file 1*, Table 1b). These domains are essential for the activity of other telomerase homologs, and have been hypothesized to be particularly important for telomerase ratcheting during translocation (*Steczkiewicz et al., 2011*). Therefore, we kept our *tc*TERT kinetics within a single turnover (i.e. insertion) regime, and, wherever possible, complemented the kinetic results with human telomerase studies to characterize the catalytic cycle of telomerase. Using this combined approach, we have elucidated the role of conserved telomerase active site residues and determined the mechanisms of fidelity and rNTP discrimination.

## Results

The TERT subunit of telomerase elongates telomeric DNA using a conserved catalytic cycle as outlined in *Figure 1A*, *Figure 1—figure supplement 3*, and here. First, telomerase anneals its RNA template to the end of telomeric DNA to form a binary complex (TERT:DNA, *Figure 1A*, state $A_1$). Next, the binary complex binds an incoming dNTP and samples for proper Watson-Crick base pairing to the RNA template (*Figure 1A*, state $B_1$). The transition between these two states represents the nucleotide binding step, measured as a dissociation constant ($K_d$). If the resulting ternary complex (TERT:DNA:dNTP) is in the proper orientation, TERT will catalyze the formation of a phosphodiester bond and extend the telomere by one nucleotide (*Figure 1A*, state $C_1$). The transition between these two states is the chemistry step, and its theoretical maximum rate with saturating nucleotide concentration is described as $k_{pol}$. Following insertion of the incoming nucleotide, telomerase will shift registry to align the active site with the next templating base (forming state $A_2$). This core catalytic cycle repeats six times, until a new telomeric repeat is added (*Figure 1A*, state $C_6$). All 18 telomerase states that are required to add one telomeric repeat are shown in *Figure 1—figure supplement 3* for reference. Importantly, as the telomerase approaches the end of its template, the DNA:RNA duplex at the 5' end begins to melt, enabling telomerase to either (1) translocate and anneal the RNA component to the newly extended telomeric repeat, thus allowing for additional repeat addition; or (2) dissociate from the telomeric DNA. The number of times that a single telomerase enzyme traverses this catalytic cycle is tightly regulated. It was recently shown telomerase becomes inactive after two repeats, but can be reactivated by the recently discovered intracellular telomerase-activating factors (iTAFs) (*Sayed et al., 2019*).

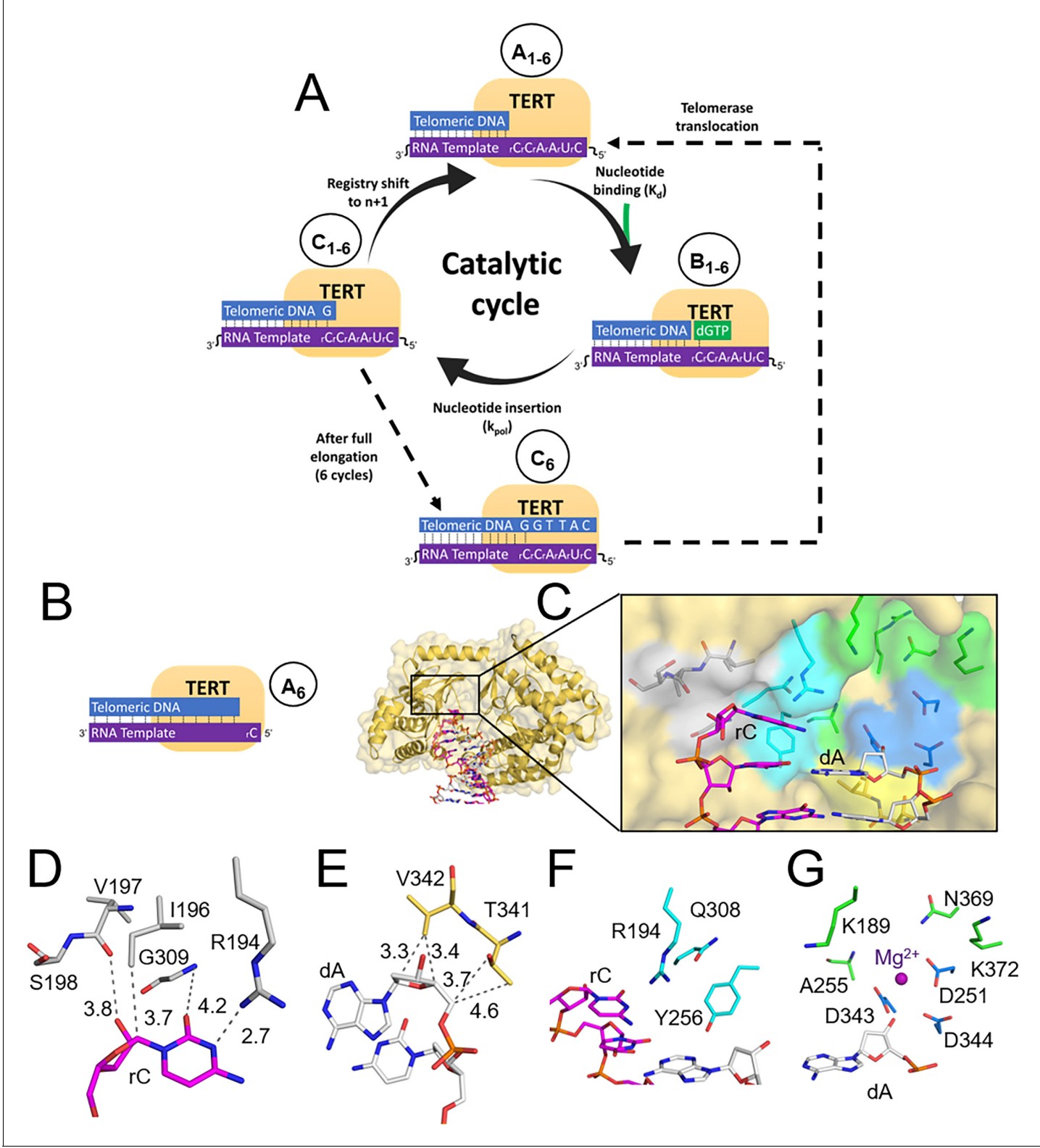

**Figure 1.** The telomerase catalytic cycle, and its first structural state. (A) Overview of the telomerase catalytic cycle. Telomerase forms a prenucleotide bound binary complex (State $A_1$). Then, it binds the incoming nucleotide triphosphate to form a ternary complex (State $B_1$), chemically links it to the telomere terminus (State $C_1$), and then shifting registry to bind the next incoming nucleotide (State $A_2$). After this cycle completes six times (State $C_6$), telomerase will either disassociate or undergo translocation (dotted line), which places it back into State $A_1$. (B) The tcTERT prenucleotide binary complex. tcTERT (pale orange cartoon and surface) encircles the DNA (white) and RNA (purple) substrate. (C) Active site pocket of the prenucleotide

*Figure 1 continued on next page*

*Figure 1 continued*

binary complex. rC binding (gray), dG binding residues (yellow), nucleoside residues (cyan), catalytic residues (blue), and triphosphate binding (green) are shown as sticks. (D) Closeup views of the rC binding residues, (E) terminal dG binding residues, (F) nucleoside binding residues, and (G) the tcTERT catalytic residues and triphosphate coordinating residues. A Mg2+ ion is shown as a purple sphere and DNA is presented as white sticks.

The online version of this article includes the following figure supplement(s) for figure 1:

**Figure supplement 1.** Alignments of several telomerase reverse transcriptase homologs.
**Figure supplement 2.** Overlay of TERT from *Tribolium castaneum* with other TERT structures.
**Figure supplement 3.** All 18 structural states involved in the extension of a telomeric repeat.

## Observing telomeric extension at the molecular level

### Pre-nucleotide binary complex

We determined how TERT engages with telomeric DNA by co-crystallizing *tc*TERT with a 16-mer RNA strand hybridized to its complementary 15-mer DNA strand to form a binary complex. This substrate mimics the initial TERT:RNA complex bound to telomeric DNA (*Figure 1A*, state $A_6$). In this orientation, an unpaired 5′ cytosine (rC) of the RNA strand acts as the templating base and a 3′ adenosine (dA) of the DNA strand serves as the primer terminus (*Figure 1B,C*). Crystals of this complex grew in a $P3_221$ space group, diffracting to 2.5 Å resolution (*Supplementary file 2*, Table 2a). The resulting structure shows TERT bound as a ring around the end of the RNA:DNA complex, with its active site positioned at the terminus of the DNA strand (*Figure 1B,C*).

Within the TERT active site, the templating RNA strand is stabilized by multiple conserved tcTERT residues (*Supplementary file 1*, Table 1a). Residues I196, V197, S198, G309, and R194 compose a pocket around the templating RNA base (*Figure 1D*). This pocket uses both polar and nonpolar interactions to stabilize the templating rC in a conformation that orients its Watson-Crick edge towards the incoming nucleotide binding site. On the opposite side, the 3′-OH of the primer terminal 3′-dA points towards the catalytic metal binding site, and side chains from T341 and V342 coordinate the deoxyribose sugar moiety of the 3′-dA with nonpolar interactions (*Figure 1E* and *Supplementary file 1*, Table 1a). This binary TERT complex also has a cavity in the active site that forms the nucleotide binding pocket. These nucleotide pocket residues can be subdivided into three categories: nucleoside coordinating, catalytic, and triphosphate interacting residues (*Figure 1F and G*). Notably, the residues that compose these three groups are 100% conserved between *h*TERT and *tc*TERT (*Supplementary file 1*, Table 1a). The nucleoside binding group is composed of residues R194, Y256, and Q308 (*Figure 1F*). These residues form a nucleoside shaped cleft directly upstream of the primer terminus of the telomeric DNA and are further characterized below. The catalytic residues include the catalytic triad: D251, D343, and D344. These residues coordinate the divalent metal ions during catalysis (*Figure 1G*). Residues K189, A255, N369, and the backbone of K372 are in position to form interactions with the triphosphate of the incoming nucleotide (*Figure 1G*). Collectively, the active site of *tc*TERT is primed for nucleotide binding, and the residues involved in binding are highly conserved with human telomerase (*Supplementary file 1*, Table 1a).

## Nucleotide bound ternary complex

We also determined the structure of TERT after binding an incoming nucleotide, but prior to catalysis (*Figure 1A*, state $B_6$). To capture the ternary complex, we utilized a non-hydrolyzable nucleotide analog 2′-deoxyguanosine-5′-[(α,β)-methyleno]triphosphate (dGpCpp). dGpCpp is identical to dGTP, except the bridging oxygen between the α and β phosphate is a carbon atom, which prevents catalysis (*Batra et al., 2006*; *Gleghorn et al., 2011*). Crystals of this complex grew in the same $P3_221$ space group and diffracted to 2.9 Å resolution (*Supplementary file 2*, Table 2a). Comparing this ternary complex to the binary state (RMSD value of 1.52 Å, *Figure 2A*) indicates minimal structural rearrangements are required for TERT to bind dGpCpp. The active site residues that compose the nucleotide binding pocket of the pre-nucleotide binary complex coordinate the incoming dGpCpp, positioning its Watson-Crick face to hydrogen bond with the templating rC (*Figure 2B,C*). Two $Mg^{+2}$ ions exhibit octahedral coordination to facilitate nucleotide binding. The catalytic metal coordinates residues D251, D343, D344, the 3′-OH of the primer terminus, and the non-bridging oxygen of the dGpCpp α-phosphate (*Figure 2D*). The nucleotide metal coordinates the side chains of D251 and D343, the backbone carbonyl of I252, and a non-bridging oxygen on the α,β, and γ

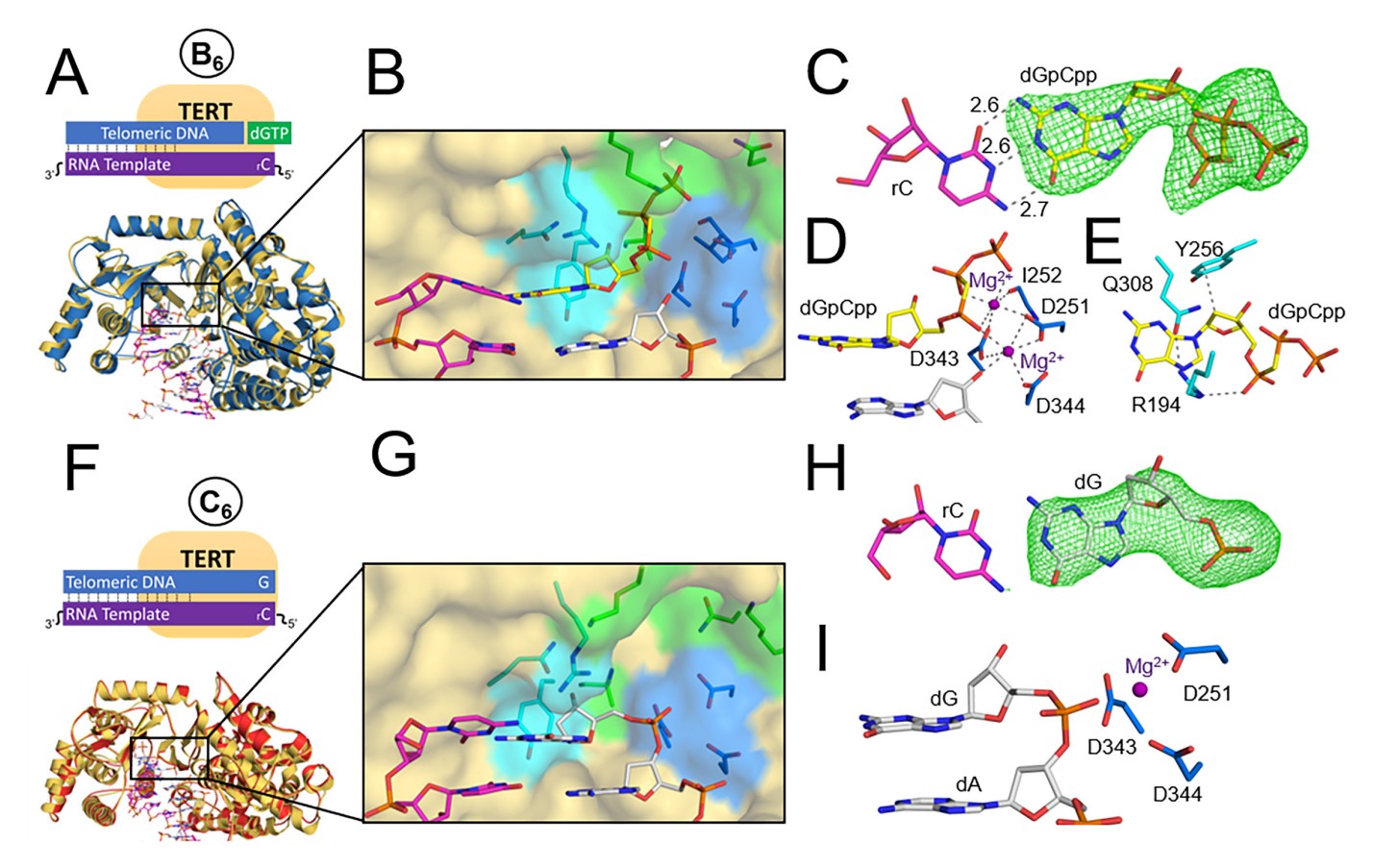

**Figure 2.** TERT ternary and product structures. (**A**) The tcTERT ternary structure, overlayed with the prenucleotide binary complex. DNA (white), RNA (purple), binary tcTERT (yellow cartoon), and ternary tcTERT (blue cartoon) are shown. (**B**) The tcTERT active site. Nucleoside residues (cyan), triphosphate residues (green), and catalytic residues (blue) are shown. The dGpCpp (yellow), DNA, and RNA are represented as sticks. (**C**) Closeup view of dGpCpp with a polder OMIT map contoured at σ = 3.0 (green mesh). (**D, E**) dGpCpp contacts are shown with Mg2+ (purple), catalytic residues (marine) and nucleoside residues (cyan) indicated. (**F**) The tcTERT product structure (red cartoon), overlaid with the prenucleotide binary complex (yellow cartoon). (**G**) An active site view of the tcTERT product structure. (**H**) A display of a polder OMIT map contoured at σ = 3.3 around the incoming dGpCpp (green mesh). (**I**) Catalytic residues (marine) coordinate the inserted dG (white).

phosphates of the incoming dGpCpp (*Figure 2D*). Nucleoside binding residue R194 remains in a similar position to where it was in the prenucleotide state, but now forms a network of contacts between residue Q308 and the α phosphate of the incoming nucleotide, stabilizing it in an orientation near other nucleoside binding residues. Y256 is positioned near the C2 position of the deoxyribose sugar portion of the nucleoside, and Q308 coordinates the nucleoside component of the dGpCpp (*Figure 2E*). As a whole, the nucleoside binding residues encircle the nucleoside component of the incoming nucleotide, and position it so that the nucleobase can base stack with the primer terminus. Overall, this ternary complex provides insight into nucleotide selection by TERT and the specific roles of active site residues during nucleotide binding.

## Product complex

To capture a product complex of telomerase, we crystallized TERT after incubation with its nucleic acid substrate and dGTP, allowing TERT to insert the nucleotide and form the final product stage of the catalytic cycle (*Figure 1A*, state $C_6$). Both globally and within the active site, we observed minimal structural changes between the prenucleotide binary complex and the product structure (RMSD value of 1.04 Å, *Figure 2F*). Electron density of the inserted dG indicates its Watson-Crick face hydrogen bonds to the Watson-Crick face of the templating rC (*Figure 2G,H*). The templating rC continues to interact with the residues that coordinated it during the other two structural states

described prior, including I196, V197, S198, G309, and R194. Neither a registry shift nor a translocation step has occurred in this structure; the TERT active site remains aligned to the terminal DNA base (*Figure 2I*). Comparing this structure to structures of the previous stages in the catalytic cycle, we observed that minimal global rearrangements are required to proceed from the binary, ternary, and product states of the catalytic cycle.

## Fidelity and sugar selectivity of TERT

Characterization of the telomerase catalytic mechanism was performed using pre-steady-state kinetics of single nucleotide insertion by *tc*TERT (*Figure 3*, *Figure 3—figure supplement 1*, *Figure 3—figure supplement 2*, and *Supplementary file 3*). These experiments determined both the $K_d$ of the

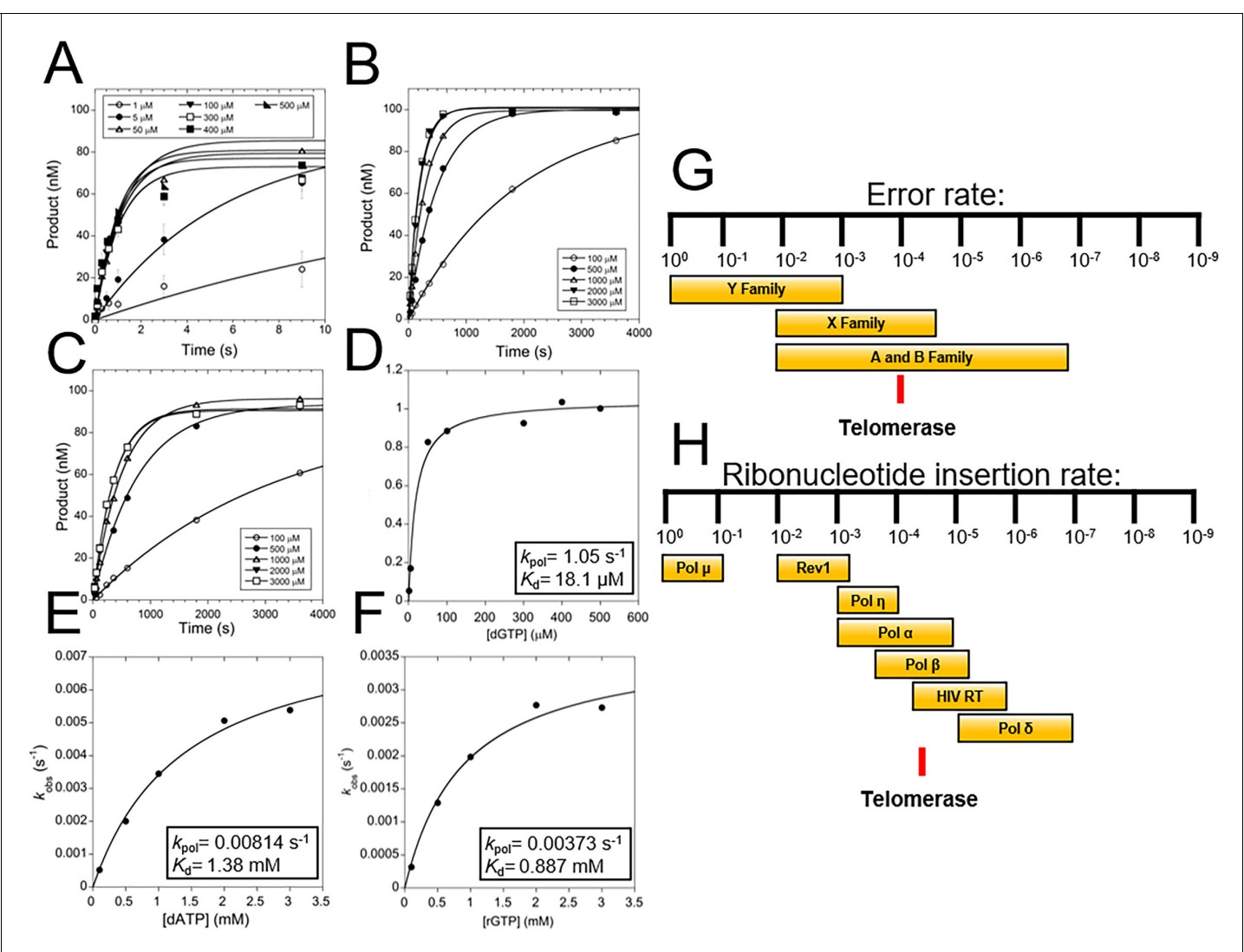

**Figure 3.** TERT fidelity and sugar discrimination. (**A**) Pre-steady-state kinetics of WT tcTERT inserting dGTP opposite rC. Data was fit to *Equation 1* (*Supplementary file 3*, Table 3a and b). Error bars represent the standard deviation of the mean. These experiments were also performed with WT tcTERT inserting dATP across from rC (**B**) and rGTP across from rC (**C**). Replots of the data and fits to *Equation 2* were performed for (**D**) dGTP across from rC, (**E**) dATP across from rC, and (**F**) rGTP across from rC. (**G**) A comparison of TERT nucleobase fidelity (red line) compared to other DNA polymerase families. (**H**) TERT's rNTP discrimination rates (red line) compared to select DNA polymerases is shown (*Brown and Suo, 2011*; *McCulloch and Kunkel, 2008*).

The online version of this article includes the following figure supplement(s) for figure 3:

**Figure supplement 1.** Pre-steady-state kinetics of additional TERT active site mutants.
**Figure supplement 2.** Complete graphs of selected TERT pre-steady-state kinetics.

incoming nucleotide and the $k_{pol}$ for nucleotide insertion by *tc*TERT, which have thus far proven unattainable for human telomerase (or any other homolog). TERT inserts the correctly matched dGTP across from a templating rC with a $k_{pol}$ of 1.05 s$^{-1}$ and a $K_d$ for the incoming dGTP of 18.1 µM (*Figure 3A,D*). Both of these values are comparable to other non-replicative DNA polymerases and the TERT $K_M$ values obtained by steady-state kinetics (*Brown et al., 2010*; *Chen et al., 2018*). We further probed the role of *tc*TERT active site residues R194 and Q308 because their role during catalysis is not clear from the structures alone and both residues protrude into the nucleotide binding pocket (*Figure 1F* and *Supplementary file 1*, Table 1a). For TERT R194A, the $k_{pol}$ decreased by 28-fold to 0.0369 s$^{-1}$, and the $K_d$ for dGTP increased ~5 fold to 93 µM (*Figure 3—figure supplement 2A,C*). With the Q308A variant, the $k_{pol}$ decreased ~60 fold to 0.30 s$^{-1}$ and the $K_d$ for dGTP increased ~2 fold to 45 µM (*Figure 3—figure supplement 2B,D*). Therefore, R194 and Q308 primarily play a role in the chemistry step rather than the nucleotide binding. As the *h*TERT homolog to R194 (R631, *Supplementary file 1*, Table 1a) is implicated in IPF, we infer that mutations at R631 likely reduce *h*TERT's $k_{pol}$, contributing to IPF pathologies (*Basel-Vanagaite et al., 2008*; *Diaz de Leon et al., 2010*).

During telomeric extension, telomerase must select between a variety of nucleic acid substrates in order to properly maintain telomeric integrity. To probe the fidelity of telomerase, we applied pre-steady-state kinetics, assessing the efficiency with which TERT inserts nucleotides during telomeric elongation. Two separate types of nucleotide selection were examined: (1) the selection of a matched dGTP over a mismatched dATP, and (2) the selection of a matched deoxyribonucletide triphosphate (dNTP) over a matched rNTP (*Figure 3B and C*, *Supplementary file 3*, Table 3a and b). We observed that for the insertion of dATP opposite a templating rC, the catalytic efficiency starkly decreased compared to dGTP insertion, both at the nucleotide binding and chemistry step. For the mismatched insertion, the $k_{pol}$ decreased 129-fold to 0.0081 s$^{-1}$ and the $K_d$ increased 76-fold to 1.3 mM (*Figure 3E*). The resulting catalytic efficiencies ($k_{pol}/K_d$) for a matched versus mismatched nucleotide insertion indicate telomerase will insert the wrong nucleotide ~1 in 10,000 nucleotide insertion events. This places telomerase at a moderate fidelity of base selection compared to other DNA polymerases (*Figure 3G*). For rNTP discrimination, the $k_{pol}$ for inserting a rGTP decreased 281-fold to 0.0037 s$^{-1}$ and the $K_d$ increased 49-fold to 0.89 mM (*Figure 3F*). This results in a nearly 14,000-fold decrease in the catalytic efficiency for the insertion of a rNTP compared to a dNTP (i.e. sugar discrimination, *Figure 3H*). Because the cellular concentrations of rNTPs are around 50-fold higher on average than dNTPs, this sugar discrimination indicates telomerase will insert a rNTP ~1 in 280 insertion events in a cellular context (see discussion) (*Traut, 1994*).

## The steric gate of telomerase

The high cellular concentration of rNTPs has resulted in most DNA polymerases evolving a structurally conserved active site residue which provides sugar discrimination by reducing the rate of rNTP insertion (*Brown and Suo, 2011*; *Cavanaugh et al., 2010*; *Nick McElhinny et al., 2010*). These residues are termed 'steric gates' because they clash with the 2'-OH of the incoming rNTP. Throughout the TERT catalytic cycle, we observed that Y256 rests in the minor groove of the DNA and is in position to clash with the 2'-OH of an incoming rNTP (*Figure 2E*). Therefore, we hypothesized this residue to be the steric gate in telomerase. To test this hypothesis, we performed pre-steady-state kinetics of rNTP insertion with the Y256A variant of TERT (*Figure 4A,B*). Compared to WT TERT, the insertion of a matched rGTP by Y256A showed a 1,490-fold increase in $k_{pol}$ to 5.5 s$^{-1}$ and a 12-fold decrease in $K_d$ to 73 µM (*Figure 4C*). The results for TERT Y256A inserting a matched dGTP were similar to that of the rGTP, with a $k_{pol}$ and $K_d$ of 6.6 s$^{-1}$ and 74 µM, respectively (*Figure 4D*). Thus, the sugar selectivity of TERT dropped from 14,000-fold between rGTP and dGTP for WT TERT to less than 2-fold for the Y256A TERT variant (*Figure 4E*). In other words, a single Y256A substitution increased rGTP insertion efficiency by 18,000-fold, abolishing almost all sugar discrimination. In a cellular environment, where rNTPs are at much higher concentrations than dNTPs, WT TERT would insert rGTP over 100-fold times less efficiently than dGTP. In contrast, TERT Y256A under cellular conditions would insert rGTP 77-fold times more efficiently than dGTP (*Figure 4F*; *Traut, 1994*).

We next verified that this ablation in sugar discrimination was due to specific changes in the TERT active site rather than global rearrangements of the enzyme. To test for structural rearrangements, we crystallized the pre-nucleotide binary state of TERT Y256A (state A$_6$), and saw minimal structural differences compared to the WT protein (*Figure 4G–J*). The position of the primer terminus was not

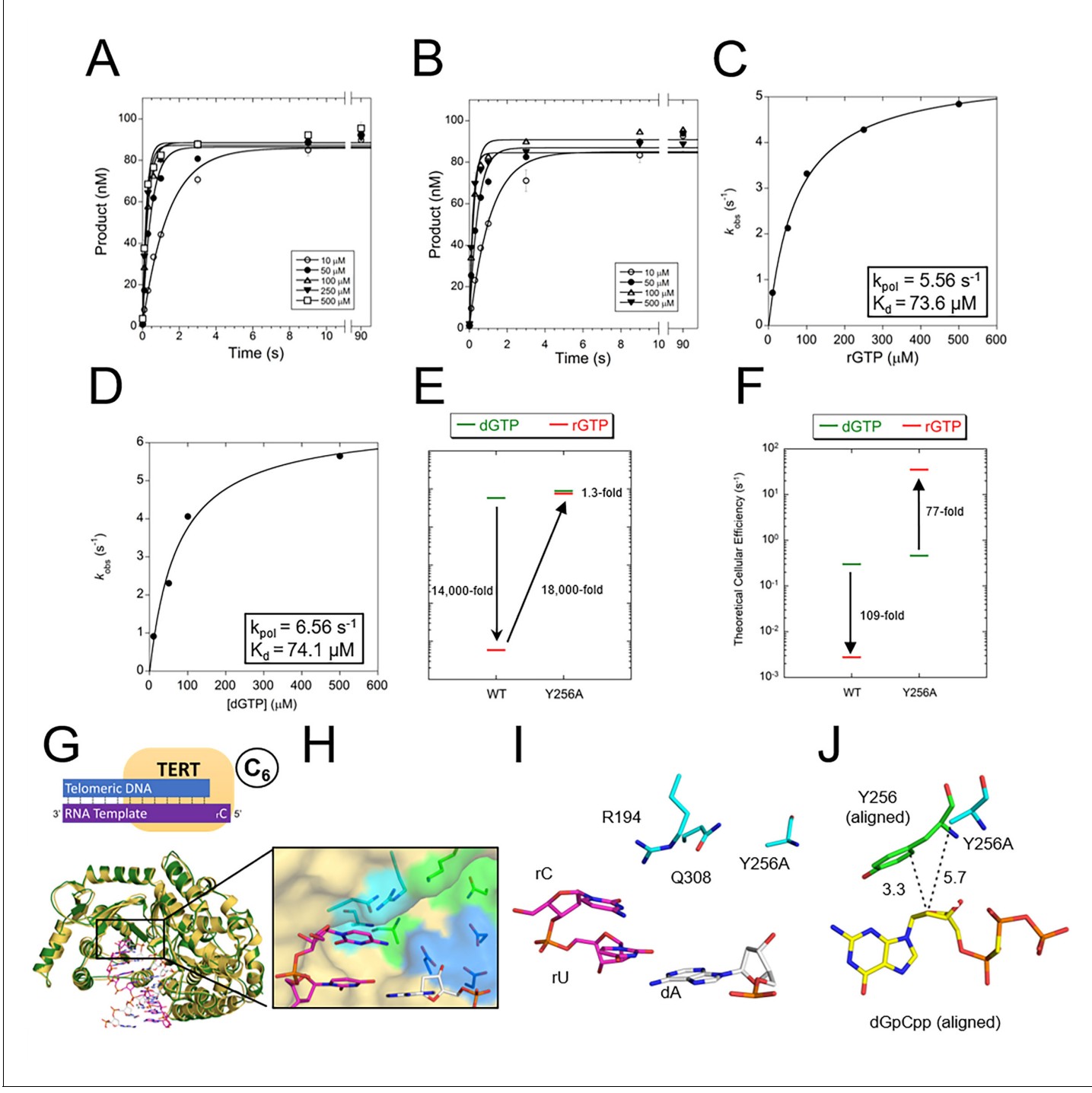

**Figure 4.** The steric gate of TERT prevents insertion of rNTPs. Pre-steady-state kinetics of tcTERT with its steric gate removed (i.e. Y256A) for both (**A**) rGTP insertion and (**B**) dGTP insertion (***Supplementary file 3***, Table 3a). Error bars represent standard deviation of the mean. These graphs were replotted and fit to ***Equation 2*** for (**C**) tcTERT Y256A inserting rGTP and (**D**) tcTERT Y256A inserting dGTP (**E**) A comparison of catalytic efficiencies (kpol/Kd) of both WT TERT and TERT Y256A for the insertion of dGTP (green) versus rGTP (red). (**F**) A comparison of TERT efficiencies adjusted for cellular nucleotide concentrations. (**G**) The prenucleotide binary complex of TERT Y256A (green) overlaid with the WT TERT prenucleotide binary complex structure (yellow). (**H, I**) The active site pocket of TERT Y256A, with DNA (white), RNA (purple), catalytic residues (blue), and nucleoside coordinating residues (cyan) shown. (**J**) The closest contacts to the C2 position of dGpCpp from the ternary structure (aligned and shown with yellow and green sticks) compared to Y256A tcTERT (cyan).

changed, and catalytic residues D251, D343, and D344 were in position to catalyze the nucleotidyl transferase reaction (*Figure 4H*). Upon closer examination of the active site pocket, the substitution of Y256 with alanine resulted in a more open nucleotide binding pocket (*Figure 4H and I*). The distance from the C2 carbon of the dGpCpp to residue 256 shifted from 3.3 Å in WT TERT to 5.7 Å in TERT Y256A, showing that the TERT Y256A has much more room to accommodate the 2'-OH (*Figure 4J*). Taken as a whole, these results indicate Y256 clashes with the 2'-OH of rNTPs to provide sugar discrimination and is the steric gate in telomerase.

To determine if human telomerase uses a similar mechanism to discriminate against rNTP insertion, we implemented human telomerase activity assays (*Xi and Cech, 2014*). In these assays, 1.5 telomeric repeats with the sequence TTAGGGTTAG were incubated with 50 µM of either all four dNTPs or all four rNTPs and purified 3xFLAG tagged human telomerase overexpressed with hTR. We performed these tests with both WT telomerase and a telomerase Y717A variant, which is the homologous residue to *tc*TERT Y256. WT telomerase showed robust primer extension in the presence of dNTPs, with ~10% of the primer extended into product over the course of 30 min (*Figure 5A,B*). Similarly, with all four dNTPs, the Y717A variant reached ~7% primer extension in the same amount of time (*Figure 5A,B*). In contrast, when we incubated WT telomerase with all four rNTPs rather than dNTPs, we observed very low (<1%) primer extension after 30 min, suggesting that human telomerase discriminates against rNTPs, similar to *tc*TERT (*Figure 5C,D*). When we performed the same telomeric extension assay with the steric gate variant (Y717A), we observed that the primer extension was increased 3-fold compared to WT *h*TERT in the presence of rNTPs (*Figure 5C,D*). These results indicate the conserved residue Y717 in *h*TERT (*Supplementary file 1*, Table 1a) is the steric gate in human telomerase. Interestingly, in both WT and Y717A telomerase, minimal insertion past the first telomeric repeat was observed, which suggests the presence of rNTPs in telomere strands may inhibit the telomerase translocation step (*Figure 5C*).

We next assessed if more subtle alterations in the nucleotide mixture would also cause inhibition of telomerase extension. By substituting one rNTP into the nucleotide mix at a time, we could determine the effects of inserting one, two, or three rNTPs inserted per repeat (with rATP, rUTP, and rGTP, respectively). In each case, telomerase processivity was reduced, with no bands evident past the second telomeric repeat (*Figure 5E*). The inhibitory effect seemed to depend on the number of rNTPs inserted per repeat, with rGTP presence showing the greatest inhibition. For the steric gate Y717A mutant telomerase, the effect of rNTPs on telomerase's processivity were much less pronounced. In the most extreme case of three rNTPs present per repeat, extension products are evident well into the second repeat, in contrast to the WT telomerase which had almost no insertion events (*Figure 5E*). We hypothesize that the rGTP insertion drastically inhibits WT telomerase because the first two insertions in the repeat are templated by rC. Therefore, the first event would need to be a rNTP insertion, followed by another rNTP insertion from a potentially unstable primer terminus. We are unable to decipher if inhibitory effects are due to the number of ribonucleotides per repeat, sequence-dependent inhibition, or a combination of both. In contrast to rGTP, telomerase is able to incorporate multiple repeats when rATP is present, albeit at a reduced efficiency compared to dNTPs (*Figure 5E*).

Within these primer extension activity assays, the effects of rNTP insertions can also be observed at the single nucleotide level. For rATP insertion with WT telomerase, we observed a buildup in substrates one nucleotide shorter than where the rNTP insertion would occur, composing ~60% of the total product formation (*Figure 5F*). This could be explained by the poor catalytic efficiency of telomerase for rATP insertion causing the enzyme to stall directly before the rATP insertion event. In contrast to the WT telomerase, the Y717A variant did not stall at the rATP insertion event. Interestingly, rNTPs inhibited translocation in both Y717A and WT telomerase, implying that telomerase possesses rNTP discrimination mechanisms beyond the level of single nucleotide incorporations (*Figure 5E,F*). The agreement between the results of the *tc*TERT and *h*TERT points towards a universal mechanism of sugar discrimination by any telomerase homolog with a tyrosine in this conserved position (*Supplementary file 1*, Table 1a).

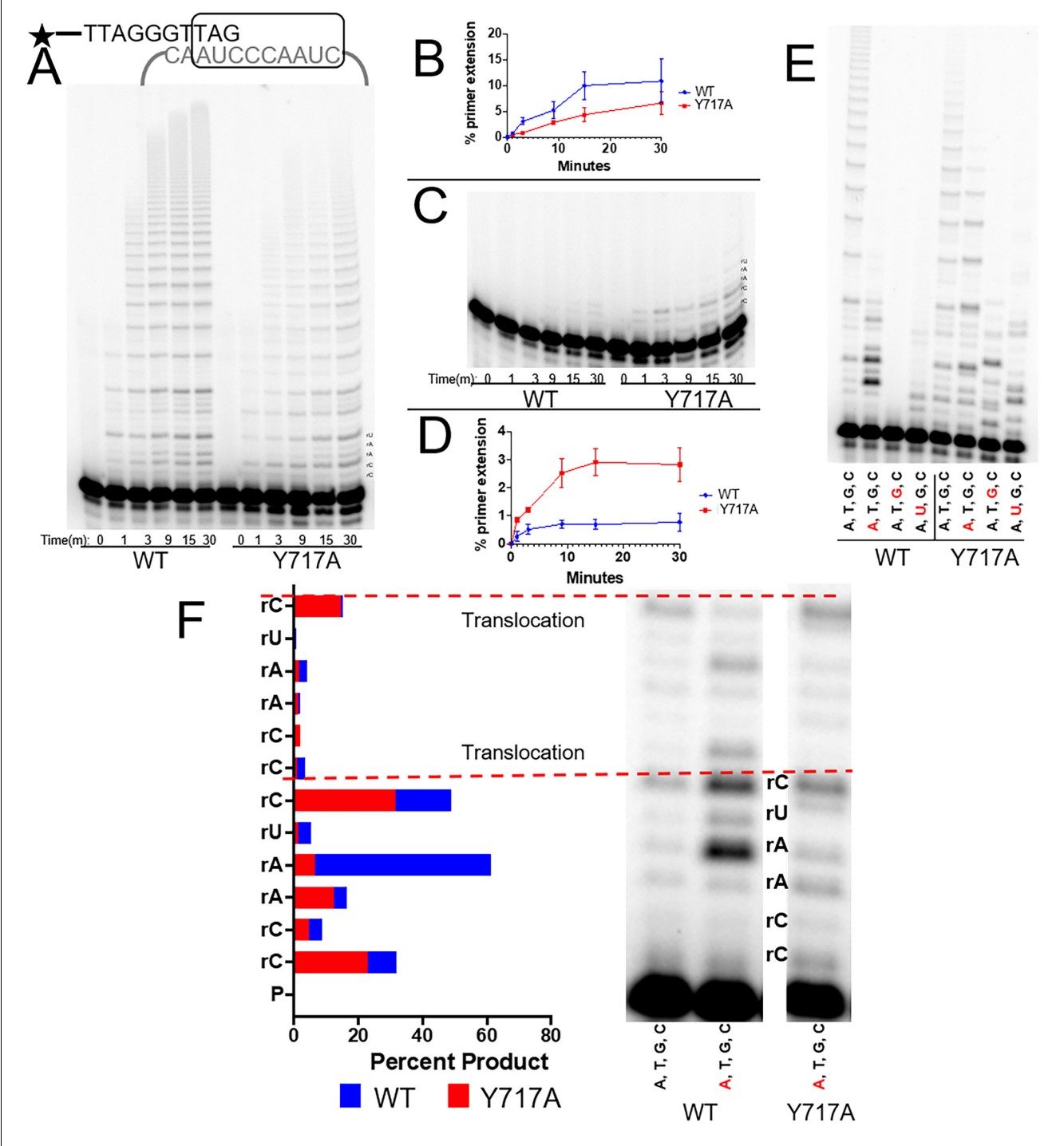

**Figure 5.** The steric gate of human telomerase. (A) Timecourse of primer extension with all four dNTPs by WT (left) and Y717A (right) human telomerase. (B) Quantification of percent primer extension, with WT shown as a blue line and the Y717A mutant shown in red. (C) Timecourse of primer extension with all four rNTPs for WT telomerase (left) and Y717A telomerase (right). (D) Results from the gel in panel C, quantified with the mutant telomerase shown in red and WT telomerase shown in blue. (E) Primer extension from both WT (left) and Y717A telomerase. Each lane contains either all four dNTPs, or three dNTP and one dNTP marked in red. (F) A close-up view of the first two telomeric repeats from selected lanes in panel E. rNTPs present in the mix are shown in red. Quantifications of each band in terms of percent product are also shown and labeled by the position of the telomerase templating base. The base marked by an asterisk is complementary to the rNTP in the solution. Error bars represent standard deviations of four biological replicates.

## Discussion

### Telomerase's catalytic cycle and fidelity

In this study, we characterized each step of the TERT catalytic cycle for single nucleotide insertion. We found that, in terms of global protein structure, minimal rearrangement is required to proceed through the catalytic cycle. This lack of rearrangement contrasts many other DNA polymerases and even HIV RT, which have been shown to undergo global shifts from an 'open' to a 'closed' state upon nucleotide binding (*Doublié et al., 1999*; *Sawaya et al., 1994*; *Schmidt et al., 2018*). Although it is unknown why the TERT catalytic core does not open and close, it may be because other complexities necessary for telomerase function limit the opening and closing from occurring, including the translocation step during repeat addition or the extensive interaction with its RNA component. Within the active site, we observed residues that encompass a cavity for the incoming nucleotide prior to binding, which then adjust to coordinate the incoming nucleotide after binding, and continue to stabilize the newly inserted base after its insertion in the product state. Many of the active site residues involved in carrying out the catalytic cycle are in similar positions as other DNA polymerases; a triad of three carboxylate containing residues such as D251, D343, and D344 in *tc*TERT is conserved in many DNA polymerases (*Steitz, 1999*). Interestingly, R194 and Q308 are in a similar structural location to R61 and Q38 of human DNA polymerase η, and both have been shown to be important in its catalytic cycle (*Biertümpfel et al., 2010*).

Our structural snapshots were complemented by kinetic studies, allowing us to understand how telomerase chooses right from wrong nucleotides; that is, selecting canonical dNTPs with correct base pairing compared to noncanonical rNTPs or mismatched base pairing (*Figure 6A*). Our experiments were carried out specifically with a single dGTP insertion using a 4 nucleotide overhang RNA template. While telomerase has been shown to exhibit moderate base and position-specific effects, our results indicate that the telomerase catalytic core generally exhibits moderate base selection fidelity, similar to that of X-family polymerases involved in DNA repair (*Chen et al., 2018*; *McCulloch and Kunkel, 2008*). Based on our kinetic values, we predict that telomerase inserts ~ 1 mismatch per each 10 kb of telomere extension. Because telomerase does not have a proofreading domain, misinsertions created by telomerase will remain as ssDNA during telomere elongation. Upon replication of the complementary strands by a DNA polymerase, the base that was a mismatch (in the context of telomerase) will become a matched base pair, and will not be a substrate for mismatch repair. Therefore, our fidelity measurement is predictive of cellular error rates in telomeric sequences. Accordingly, our predicted error rate agrees with telomeric error rates observed using telomere sequencing (*Lee et al., 2018*). While the downstream consequences of telomeric mismatches have not been studied in a biological context to our knowledge, they likely would disrupt G-quadruplex stability and inhibit shelterin protein binding, as both of these phenomena are dependent on DNA sequence (*Figure 6B*; *Burge et al., 2006*; *de Lange, 2005*).

### Telomeric ribonucleotides

DNA polymerases insert millions of rNTPs into the genome during replication, because of a large disparity in nucleotide concentrations (rNTPs are ~50 fold more abundant in cells than dNTPs) (*Traut, 1994*). Telomerase also must select against this disparity; although telomerase is canonically thought to elongate telomeres with only dNTPs, our kinetics imply this is not the case. Instead, we predict that for every 10 kb of telomere extension, telomerase inserts ~ 40 rNTPs, which represents selectivity comparable to DNA polymerase β and DNA polymerase δ (*Brown and Suo, 2011*; *Cavanaugh et al., 2010*; *Nick McElhinny et al., 2010*). However, it is unknown whether ribonucleotides persist in telomeres, their biological consequences, and if they are addressed with ribonucleotide excision repair (RER), similar to other genomic ribonucleotides (*Sparks et al., 2012*). In our experiments with human telomerase, we found that even with increased rates of ribonucleotide insertion at the single nucleotide insertion level, telomere elongation was reduced via an inhibition of the translocation step (*Figure 5E,F*). This reduction was evident even with a single rNTP present in a telomeric repeat. Furthermore, previous studies have found telomeric substrates containing ribonucleotides can prevent or reduce extension of the first repeat depending on the number and position of ribonucleotides present in the DNA template (*Collins and Greider, 1995*). It is possible that telomerase pauses after inserting ribonucleotides to provide an opportunity for an extrinsic

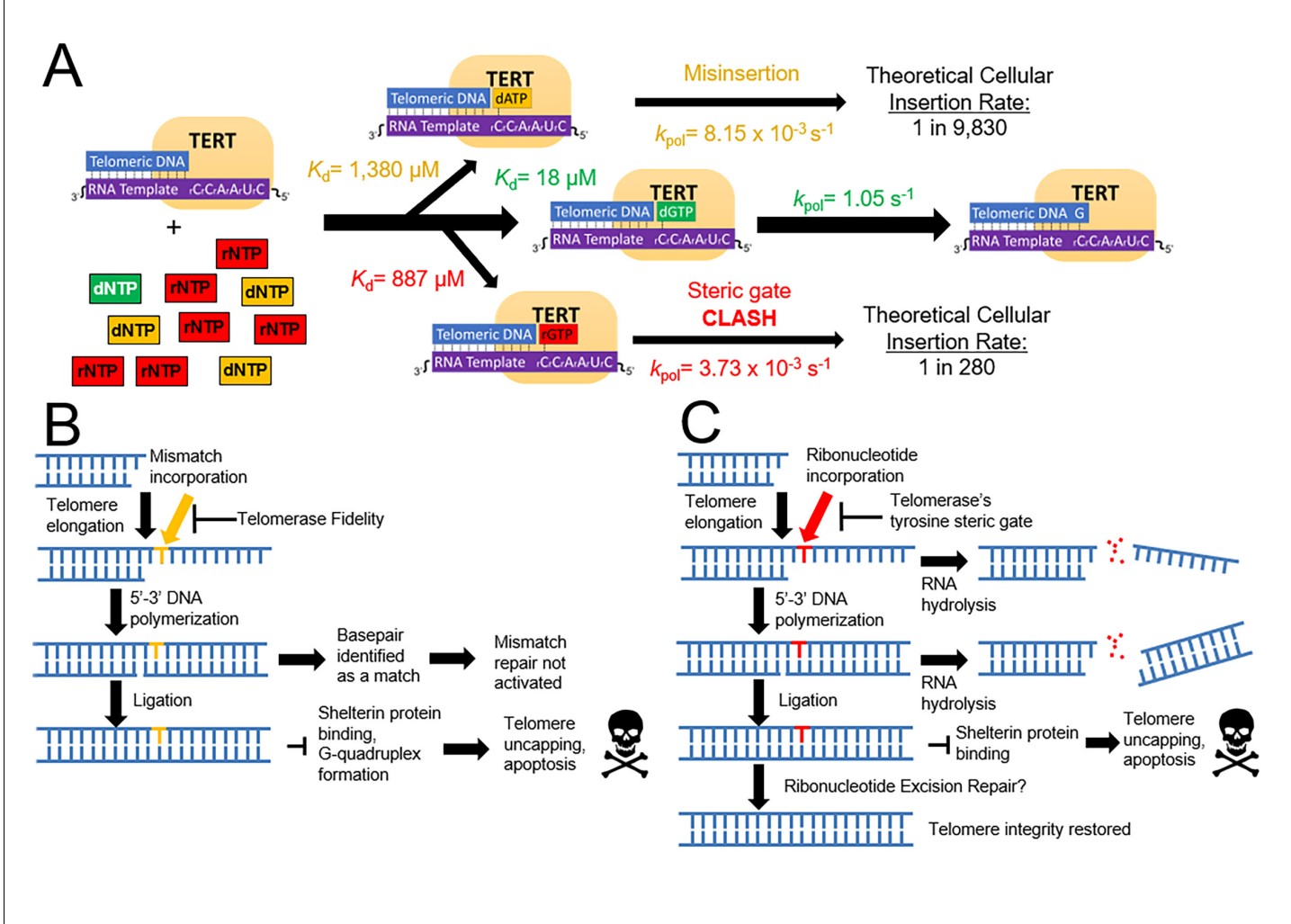

**Figure 6.** A model for how telomerase preserves telomeric integrity and biological implications. (**A**) The telomerase catalytic cycle, with matched dNTP (green), mismatched dNTPs (orange), and matched rNTPs (red). For each nucleotide path, kinetic parameters are labeled for each step and the insertion probability. (**B**) Downstream consequences of mismatch insertion by telomerase may include evasion of mismatch repair, reduced binding affinity for shelterin proteins, and altered stability of telomeric G quadruplexes. (**C**) Eventual consequences of rNTP insertion by telomerase. After insertion into telomeres, ribonucleotides could cause harsh consequences via hydrolysis, or disrupting telomere capping and stability. Ribonucleotides may also be removed by ribonucleotide excision repair.

proofreader or RER to remove the ribonucleotide before continuing telomeric elongation. Telomerase has previously been observed to halt telomeric extension after the insertion of other noncanonical nucleotides, particularly in the case of 8-oxodeoxyguanine triphosphate (8-oxodGTP) insertion (*Fouquerel et al., 2016*). Therefore, telomerase may stall on noncanonical nucleotides, such as 8-oxodGTP, for similar proofreading reasons.

Telomerase discrimination against rNTP insertion is important because ribonucleotides can cause multiple downstream problems for telomeric stability. These problems can arise from both the direct reduction of telomere length and from altering telomere structure. Because RNA is more vulnerable to hydrolysis than DNA, inserted ribonucleotides are more likely to hydrolyze, which would cause a telomeric single strand break (*Baumann and Cech, 2001*; *de Lange, 2005*; *Li and Breaker, 1999*). If the strand break occurred before the complementary strand of the telomere was copied and ligated to the genome, the elongating single strand would be released, and the entire telomere elongation process would need to be restarted (*Figure 6C*). Alternately, the inserted ribonucleotide may not hydrolyze, but instead persist within telomeres. The continued presence of ribonucleotides in telomeres may cause several structural aberrations, including altering G-quadruplex stability and

disrupting shelterin protein binding (*Fay et al., 2017*; *Nowotny et al., 2005*). If shelterin proteins are not able to bind to telomeres, telomere ends may be recognized as DNA damage, activating double strand break repair and causing disastrous biological consequences.

Subtle alterations to the telomeric nucleotides have previously been shown to cause telomeric disruption in a cellular context (*Fouquerel et al., 2016*; *Mender et al., 2015*; *Stefl et al., 2001*). Even changes of only one atom in a nucleotide, including replacing an oxygen of a guanine with a sulfur to form therapeutic 6-thioguanine nucleotides or the adduction of an extra oxygen onto guanine to form 8-oxoguanine nucleotides, have significant biological consequences in the context of telomerase. Therefore, in order to prevent this telomeric disruption, the telomerase active site appears to have evolved a high degree of stringency towards noncanonical nucleotides, including both rNTPs and mismatched dNTPs. This stringency was evident by the reduced telomere elongation efficiency with every variant tested; other mutations could also be identified with this system that show increases in telomere elongation efficiency. Our structural characterization of the telomerase catalytic cycle allowed us to efficiently modify the stringent active site of telomerase, generating a human telomerase variant that readily inserts rNTPs. The applications shown here highlight the potential of combining model TERTs with complementary human telomerase studies to further probe the telomerase catalytic mechanism and screen future telomerase-targeting therapeutics.

## Materials and methods

**Key resources table**

| Reagent type (species) or resource | Designation | Source or reference | Identifiers | Additional information |
|---|---|---|---|---|
| Gene (*Tribolium castaneum*) | tcTERT | GenScript | | |
| Gene (*Homo sapiens*) | hTR | Gift from Dr. Tom Cech | | |
| Gene (*Homo sapiens*) | hTERT | Gift from Dr. Tom Cech | | |
| Strain, strain background (*Escherichia coli*) | One Shot BL21(DE3)pLysS Chemically Competent *E. coli* | Invitrogen | Cat# C606010 | |
| Strain, strain background (*Escherichia coli*) | One Shot TOP10 Chemically Competent *E. coli* | Invitrogen | Cat# C606010 | |
| Cell line (*Homo sapiens*) | HEK 293 T Cells, female | ATCC | RRID:CVCL_0063 | Cells were acquired from ATCC, and have not since been tested for mycoplasma, as they were used for protein generation not biological assays |
| Transfected construct (*Homo sapiens*) | pSUPER-hTR | Gift from Dr. Tom Cech | N/A | |
| Transfected construct (*Homo sapiens*) | pVan107 3X FLAG hTERT | Gift from Dr. Tom Cech | N/A | |
| Recombinant DNA reagent | pET-28a(+) with *Tribolium castaneum* TERT | Genscript | N/A | |
| Sequence-based reagent | Primer for telomerase activity assays | IDT | N/A | 5'-GGTCAGGT CAGGTCA-3' |

*Continued on next page*

*Continued*

| Reagent type (species) or resource | Designation | Source or reference | Identifiers | Additional information |
|---|---|---|---|---|
| Sequence-based reagent | RNA template for *tc*TERT kinetics | IDT | N/A | 5'-rCrUrGrArCrCr UrGACCUGACC-3' |
| Sequence-based reagent | DNA primer for *tc*TERT kinetics | IDT | N/A | 5'-/6-FAM/CCAG CCAGGTCAG-3' |
| Sequence-based reagent | RNA template for *tc*TERT crystallogarphy | IDT | N/A | 5'- rUrGrArCrCrUrGr ArCrCrUrGrG rCrUrGrG-3' |
| Sequence-based reagent | DNA primer for *tc*TERT crystallogarphy | IDT | N/A | 5'-GGTTAGGGT TAGGGTTAG-3' |
| Peptide, recombinant protein | T4 polynucleotide kinase | NEB | Cat# M0201S | |
| Peptide, recombinant protein | 3X FLAG Peptide | Sigma Aldrich | Cat# F4799-4MG | |
| Chemical compound, drug | 2'-deoxyguanosine-5'-[(α,β)-methyleno] triphosphate (dGpCpp) | Jena Biosciences | Cat# NU-431S | |
| Chemical compound, drug | γ−32P ATP | Perkin-Elmer | Cat# BLU0 02Z250UC | |
| Chemical compound, drug | 2-methyl-2, 4-pentanediol | Hampton Research | Cat# HR2-627 | |
| Software, algorithm | COOT | *Emsley and Cowtan (2004)* | https://www2. mrc-lmb.cam. ac.uk/personal/ pemsley/coot/ | RRID:SCR_014222 |
| Software, algorithm | Phenix | *Adams et al., 2010* | https://www. phenix-online.org/ | RRID:SCR_014224 |
| Software, algorithm | XDS | *Kabsch (2010)* | http://xds. mpimf-heidelberg. mpg.de/ | RRID:SCR_015652 |
| Software, algorithm | MolProbity | *Chen et al. (2010)* | http://molprobity. biochem.duke.edu/ | RRID:SCR_014226 |
| Software, algorithm | ImageJ | *Schneider et al., 2012* | https://imagej. nih.gov/ij/ | RRID:SCR_003070 |
| Software, algorithm | Kaleidagraph | Synergy Software | http://www. synergy.com/ wordpress_ 650164087/ kaleidagraph/ | RRID:SCR_014980 |
| Software, algorithm | ImageQuant TL v8.1 | GE Healthcare Life Sciences | http://www. gelifesciences. com/en/us | RRID:SCR_014246 |
| Software, algorithm | PyMol | Schrödinger LLC | https://pymol. org/2/ | RRID:SCR_000305 |
| Other | Large scale expresion (LEX-48) bioreactor | Epiphyte | https://www. epiphyte3.com/LEX | |
| Other | HisTrap HP 5 mL column | GE healthcare Life Sciences | Cat# 17524801 | |

*Continued on next page*

*Continued*

| Reagent type (species) or resource | Designation | Source or reference | Identifiers | Additional information |
|---|---|---|---|---|
| Other | POROS HS strong cation ion exchange resin | Thermo scientific | Cat# 1335906 | |
| Other | G-25 spin columns | GE Healthcare Life Sciences | Cat #27532501 | |
| Other | Sephacryl 16/60 S-200 HR Size Exclusion Chromatography column | GE Healthcare Life Sciences | Cat # 17116601 | |
| Other | ANTI-FLAG M2 affinity gel agarose beads | Sigma Aldrich | Cat #A2220 | |

## Nucleic acid sequences

To generate crystal structures of the TERT catalytic cycle, the following DNA sequences were utilized for all crystallization experiments: DNA primer of 5'-GGTCAGGTCAGGTCA-3' and the RNA template sequence 5'-rCrUrGrArCrCrUrGACCUGACC-3'. For kinetic studies, we utilized a DNA primer with a 5' label of 6-carboxyfluorescein (6-FAM), and the DNA sequence of 5'- CCAGCCAGGTCAG-3'. The RNA template used in kinetic reactions contained the sequence 5'- rUrGrArCrCrUrGrArCrC rUrGrGrCrUrGrG-3' and was not labeled. In each case, the oligonucleotides were resuspended in molecular biology grade water, and the concentration was calculated from their absorbance at 260 nm as measured on a NanoDrop microvolume spectrophotometer. Nucleic acid substrates for crystallography were annealed at an equimolar ratio, but nucleic acid substrates for our kinetic studies were annealed at a 1:1.2 molar ratio of labeled to unlabeled primer. We used a thermocycler to anneal all nucleic acid substrates, heating them to 90°C for 2 min before cooling to 4°C at a rate of 0.1°C per second.

## Expression and purification of *tc*TERT

We used previously published methods for *tc*TERT expression and purification, but implemented several modifications (*Gillis et al., 2008*). Briefly, we grew *tc*TERT in BL-21(DE3)pLysS cells using an Epiphyte3 LEX bioreactor at 37°C until they reached an OD600 of 0.6–0.8, after which the temperature was dropped to 30°C for 4–5 hr of protein production. Cells were harvested via centrifugation at 4000 x g until lysis. For TERT purification, we used buffers containing 0.75 M KCl and 10% glycerol for the capture step on Ni-NTA columns (GE Healthcare), and then further purified our sample with cation exchange on a POROS HS column (Thermo Fisher), using a salt gradient of 0.5 M KCl to 1.5 M KCl. Then, we cleaved the hexahistadine tag with Tobacco etch virus protease before purifying the cut tag from the protein with another run on our Ni-NTA columns. Finally, we used a slightly different buffer for the our size exclusion chromatography (Sephacryl S-200 16/60, GE Helathcare), containing 50 mM Tris-HCl, pH 7.5, 10% glycerol, 0.8 M KCl and 1 mM Tris(2-carboxyethyl)phosphine (TCEP). Resultant *tc*TERT was concentrated down to 18 mg mL$^{-1}$ prior to crystallography, and stored at 4°C (*Gillis et al., 2008*).

## Crystallization of *tc*TERT

Prior to crystallization, we complexed *tc*TERT with its nucleic acid substrate by mixing them at a 1:1.2 ratio of protein to DNA. To increase protein solubility, we included 520 mM KCl when preparing to mix *tc*TERT with its nucleic acid substrate. We then used sitting drop vapor diffusion to grow binary complex crystals in conditions containing 11% isopropanol, 0.1 M KCl, 25 mM MgCl$_2$, and 50 mM sodium cacodylate pH 6.5. Volume ratios for the optimal crystal growth were optimized to 2.3 μL *tc*TERT binary complex crystals + 1.7 μL of our crystallization condition, to make 4 μL total. For the ternary complex crystals, we used the same conditions, but included 0.69 mM dGpCpp (Jena Biosciences), the next matched incoming nucleotide in the sequence. Finally, for the product complex, we formed a DNA strand one nucleotide longer by incubating 2.5 mM dGTP with *tc*TERT and

its nucleic acid substrate, allowing the reaction to occur at 22°C for 30 min prior to setting up crystallization drops. In all cases, crystals were transferred to a cryosolution containing 80% reservoir solution and 20% 2-methyl-2,4-pentanediol by volume before flash cooling them in liquid nitrogen.

## Data collection and refinement

All datasets were collected at a wavelength of 1.00 Å, using the 4.2.2 synchrotron beamline at the Advanced Light Source of the Ernest Orlando Lawrence Berkeley National Laboratory. Datasets were indexed and scaled using XDS (RRID:SCR_015652) (*Kabsch, 2010*; *Winn et al., 2011*). Initial models were generated using molecular replacement in PHENIX (RRID:SCR_014224), using a previously published *tc*TERT structure with an alternate substrate, PDB code 3KYL (*Adams et al., 2010*; *Mitchell et al., 2010*). After a solution was found, all DNA and RNA bases were built in the pre-nucleotide complex, and the resultant structure was then used for further molecular replacements other structures. Model building was accomplished with Coot (RRID:SCR_014222) and validated with MolProbity (RRID:SCR_014226) (*Chen et al., 2010*; *Emsley and Cowtan, 2004*). For structures ≥ 3 Å resolution, both secondary structure restraints and torsional restraints from the prenucleotide binary structure were used to prevent overmodeling. All refinements were done using PHENIX, and figures were generated using PyMOL (RRID:SCR_000305, Schrödinger LLC). For each of the structures, Ramachandran analysis revealed a minimum of 100% of non-glycine residues occupied allowed regions and at least 93% occupied favored regions.

## Pre-steady-state kinetic characterization of *tc*TERT

Pre-steady-state kinetic parameters of *tc*TERT were obtained using established pre-steady-state kinetics protocols for DNA polymerases, also known as single turnover kinetics (*Beard et al., 2014*; *Powers and Washington, 2017*). Briefly, we preincubated 2 µM *tc*TERT with 200 nM annealed DNA:RNA hybrid substrate, with a 6-FAM label on the 5' end of the DNA component. We then used a KinTek RQF-3 (a rapid quench-flow instrument) to mix equal ratios of the incoming nucleotide triphosphate of interest and 10 mM MgCl$_2$ with the existing mix of *tc*TERT and its DNA:RNA hybrid substrate. Reactions were run at 37°C and quenched at various timepoints (ranging from 10 ms to 700 s) with 100 mM EDTA pH 7.5. In each case, the conditions used for each reaction were: 25 mM TRIS pH 7.5, 0.05 mg mL$^{-1}$ Bovine Serum Albumin, 1 mM dithiothreitol, 10% glycerol, 200 mM KCl, 1 µM *tc*TERT, 100 nM annealed DNA:RNA hybrid substrate, and varying concentrations of the nucleotide triphosphate of interest. The samples were transferred to a DNA gel loading buffer, containing 100 mM EDTA, 80% deionized formamide, 0.25 mg ml$^{-1}$ bromophenol blue and 0.25 mg ml$^{-1}$ xylene cyanol. For the generation of data sets that had a minimum time point of 12 s or greater, a LabDoctor heating block was used in lieu of the KinTec RQF-3, and quenching was accomplished using a solution of DNA gel loading buffer. These mixes were then incubated at 95°C for 5 mins and loaded onto a 21% denaturing polyacrylamide gel. These gels were run at 700 V, 60 A, and 30 W at 30°C in order to separate the reaction product from its substrate.

Gels were scanned and imaged using a GE Typhoon FLA 9500 imager, and the ratios of product to substrate were quantified using ImageJ (RRID:SCR_003070) (*Schneider et al., 2012*). Means and standard deviations were taken from at least three technical replicates were calculated and graphed using KaleidaGraph (RRID:SCR_014980). Plots of product formation over time were fit to the exponential *Equation 1* to determine k$_{obs}$ values:

$$[P] = A\left(1 - e^{-k_{obs}t}\right) \tag{1}$$

[P] is the concentration of the product, A is the target engagement (amplitude), and t is the reaction time. After k$_{obs}$ values were determined for multiple nucleotide triphosphate concentrations, the data was replotted to compare k$_{obs}$ to concentration of nucleotide triphosphate, and fit to *Equation 2*:

$$k_{obs} = \frac{k_{pol}[NTP]}{K_d + [NTP]} \tag{2}$$

$k_{pol}$ represents the theoretical maximum value of $k_{obs}$, and [NTP] represents the concentration of the nucleotide of interest.

## Telomerase expression

HEK293T cells were used to overexpress hTR and 3 × FLAG tagged human telomerase reverse transcriptase (hTERT) genes in pSUPER-hTR and pVan107, respectively. Cells were grown to 90% confluency in Dulbecco's modified Eagle's medium (Gibco) supplemented with 10% High Quality FBS (Hyclone) and 1% penicillin-streptomycin (Corning) at 37°C and 5% CO2. Cells were transfected with 10 µg of pSUPER-hTR plasmid and 2.5 µg of pVan107 hTERT plasmid diluted in 625 µl of Opti-MEM (Gibco) using 25 µl of Lipofectamine 2000 (ThermoFisher) diluted in 625 µl of Opti-MEM. Cells were cultured for 48 hr post-transfection, and then were trypsinized and washed with phosphate-buffered saline and lysed in CHAPS lysis buffer buffer (10 mM Tris-HCl, 1 mM MgCl$_2$, 1 mM EDTA, 0.5% CHAPS, 10% glycerol, 5 mM β-mercaptoethanol, 120 U RNasin Plus (Promega), 1 µg/ml each of pepstatin, aprotinin, leupeptin and chymostatin, and 1 mM AEBSF) for 30 min at 4°C. Cell lysate supernatant was then flash frozen and stored at −80°C.

## Telomerase purification

Telomerase was purified via the 3xFLAG tag on hTERT encoded pVan107 using ANTI-FLAG M2 affinity gel agarose beads (Sigma Aldrich), as described previously with some modification (*Fouquerel et al., 2016*). An 80 µL bead slurry (per T75 flask) was washed three times with 10 volumes of 1X human telomerase buffer in 30% glycerol with 1 min centrifugation steps at 3500 r.p.m. at 4°C. The bead slurry was added to the lysate and nutated for 4–6 hr at 4°C. The beads were harvest by 1 min centrifugation at 3500 r.p.m, and washed 3X with 1X human telomerase buffer with 30% glycerol. Telomerase was eluted from the beads using 2x the bead volume of 250 µg/mL 3X FLAG peptide (Sigma Aldrich) in 1X telomerase buffer with 150 mM KCl. The bead slurry was nutated for 30 min at 4°C. The eluted telomerase was collected using Mini Bio-Spin Chromatography columns (Bio-Rad). Samples were flash frozen and stored a −80°C.

## 32P-end-labeling of DNA primers

50 pmol of PAGE purified DNA primer GGTTAGGGTTAGGGTTAG (IDT) was labeled with $\gamma-32P$ ATP (Perkin Elmer) using T4 polynucleotide kinase (NEB) in 1X PNK Buffer (70 mM Tris-HCl, pH 7.6, 10 mM MgCL2, 5 mM DTT) in a 20 uL reaction volume. The reaction was incubated for 1 hr at 37°C followed by heat inactivation at 65°C for 20 min. G-25 spin columns (GE Healthcare) were used to purify the end labeled primer.

## Telomerase activity assay

The telomerase assay was as previously described. Reactions contained 1x human telomerase buffer, 5 nM of 32P-end-labeled primer and 50 µM dNTP or rNTP mix as indicated in the figure legends. Each reaction was performed with four biological replicates. The reactions were started by the addition of 3 µL of immunopurified telomerase eluent, incubated at 37°C for a specified time course, then terminated with 2 µL of 0.5 mM EDTA and heat inactivated at 65°C for 20 min. An equal volume of loading buffer (94% formamide, 0.1 × Tris borate-EDTA [TBE], 0.1% bromophenol blue, 0.1% xylene cyanol) was added to the reaction eluent from the G-25 spin column. The samples were heat denatured for 10 min at 100°C and loaded onto a 14% denaturing acrylamide gel (7M urea, 1x TBE) and electrophoresed for 90 min at constant 38W. Samples were imaged using a Typhoon phosphorimager (GE Healthcare). Percent primer extension was quantitated using ImageQuant (RRID:SCR_014246).

## Crystallographic statistics

Resolution of our crystal structures was determined using correlation coefficients (CC$_{1/2}$), with the highest resolution shell containing a CC$_{1/2}$ value of greater than 0.3 (*Supplementary file 2*, Table 2a). During refinement, the statistics of R$_{work}$ and R$_{free}$, as calculated by PHENIX, were used to identify a model's fit to electron density. See *Supplementary file 2*, Table 2a for more details of these parameters for each dataset.

## Data resources

Accession numbers for models reported are PDB: 6USO, 6USP, 6USQ, and 6USR.

## Acknowledgements

We thank Jay Nix (Molecular Biology Consortium 4.2.2 beamline at Advanced Light Source) for aid in remote data collection and help with data analysis. This research used resources of the Advanced Light Source, which is a Department of Energy Office of Science user facility under Contract DE-AC02-05CH11231. We thank Amy Whitaker (University of Kansas Medical Center) for helpful discussion and assistance with the manuscript preparation, and Scott Lovell (University of Kansas) for his advice and help with crystal optimization. This research was supported by the National Institute of General Medical Science [R35-GM128562 to BDF, MAS, GAW, THK], the National Institute of Environmental Health Sciences [R35-ES030396 to PLO, SLS, SAJ], the Madison and Lila Self Graduate Fellowship [to MAS], and a NIH Clinical and Translational Science Award grant (UL1 TR002366) awarded to the University of Kansas Medical Center.

## Additional information

### Funding

| Funder | Grant reference number | Author |
| --- | --- | --- |
| National Institute of General Medical Sciences | R35-ES030396 | Patricia L Opresko<br>Samantha L Sanford<br>Samuel A Johnson |
| National Institute of Environmental Health Sciences | R35-GM128562 | Bret D Freudenthal<br>Matthew A Schaich<br>Griffin A Welfer<br>Thu H Khoang |
| University of Kansas | Madison and Lila Self Graduate Fellowship | Matthew A Schaich |

The funders had no role in study design, data collection and interpretation, or the decision to submit the work for publication.

### Author contributions

Matthew A Schaich, Conceptualization, Data curation, Formal analysis, Investigation, Writing - original draft, Writing - review and editing; Samantha L Sanford, Griffin A Welfer, Samuel A Johnson, Thu H Khoang, Data curation; Patricia L Opresko, Supervision, Writing - review and editing; Bret D Freudenthal, Conceptualization, Supervision, Funding acquisition, Writing - original draft, Writing - review and editing

### Author ORCIDs

Matthew A Schaich https://orcid.org/0000-0001-6771-5623
Bret D Freudenthal https://orcid.org/0000-0003-1449-4710

### Decision letter and Author response

Decision letter https://doi.org/10.7554/eLife.55438.sa1
Author response https://doi.org/10.7554/eLife.55438.sa2

## Additional files

### Supplementary files

• Supplementary file 1. Table 1a. The conservation and function of TERT active site residues. Table 1b. Telomerase architecture of five different TERTs, including *Homo sapiens, Tetrahymena thermophila,Saccharomyces cerevisiae* and *Tribolium castaneum*. Data from this table used from: (*Garforth et al., 2006*; *Gillis et al., 2008*; *Jiang et al., 2018*; *Nguyen et al., 2018*; *Niederer and Zappulla, 2015*).[1]TR = Telomerase RNA component*TEN = N terminal domain*TRBD = Telomerase RNA binding domain*RT = Reverse transcriptase domain*CTE = C terminal extension*IFD = Insertion in fingers domain[2] This IFD is truncated compared to other TERTs.

- Supplementary file 2. Table 2a. Data collection and refinement statistics for *tc*TERT structures.

- Supplementary file 3. Table 3a. Kinetic parameters and errors for *tc*TERT pre-steady-state kinetics of single nucleotide incorporation. Table 3b. Full kinetic parameters of each curve used to generate K$_{obs}$ and target engagement values in *tc*TERT pre-steady-state kinetics

- Transparent reporting form

### Data availability

Diffraction data have been deposited in PDB under the accession code 6USO, 6USP, 6USQ, and 6US.

The following datasets were generated:

| Author(s) | Year | Dataset title | Dataset URL | Database and Identifier |
|---|---|---|---|---|
| Freudenthal BD, Schaich MA | 2020 | Telomerase Reverse Transcriptase prenucleotide binary complex, TERT:DNA | https://www.rcsb.org/structure/6USO | RCSB Protein Data Bank, 6USO |
| Freudenthal BD, Schaich MA | 2020 | Telomerase Reverse Transcriptase ternary complex, TERT:DNA: dGpCpp | https://www.rcsb.org/structure/6USR | RCSB Protein Data Bank, 6USR |
| Freudenthal BD, Schaich MA | 2020 | Telomerase Reverse Transcriptase product complex, TERT:DNA | https://www.rcsb.org/structure/6USP | RCSB Protein Data Bank, 6USP |
| Freudenthal BD, Schaich MA, Khoang TH | 2020 | Telomerase Reverse Transcriptase binary complex with Y256A mutation, TERT:DNA | https://www.rcsb.org/structure/6USQ | RCSB Protein Data Bank, 6USQ |

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
