## [Decision Letter]

**Acceptance summary:**

The reviewers and the reviewing editor agreed that this work is significant because it is not known how telomerase selects the correct nucleotide for insertion into the telomeric repeat from a pool of nucleotide with various sugars and base pairing properties. This work is an important advance defining (at least in vitro) nucleotide selection by telomerase through the first high resolution crystal structures of *Tribolium castaneum* telomerase reverse transcriptase (TERT) throughout its catalytic cycle and characterization of the Tribolium and human TERT kinetics. The authors solved the three structures of Tribolium TERT in complex with a DNA/RNA duplex with 1 nucleotide (nt) overhang, the same duplex with a non-hydrolyzable analog of dGTP and a product DNA/RNA duplex. The most important new finding was that there is a steric gate that allows TERT to discriminate dNTPs from rNTPs. The authors show that this residue (Y717) serves similar functions in human telomerase.

**Decision letter after peer review:**

Thank you for submitting your article "Mechanisms of nucleotide selection by telomerase" for consideration by *eLife*. Your article has been reviewed by three peer reviewers, and the evaluation has been overseen by a Reviewing Editor and Cynthia Wolberger as the Senior Editor. The reviewers have opted to remain anonymous.

The reviewers have discussed the reviews with one another and the Reviewing Editor has drafted this decision to help you prepare a revised submission.

Summary:

The reviewers and the reviewing editor agreed that this work is significant because it is not known how telomerase selects the correct nucleotide for insertion into the telomeric repeat from a pool of nucleotide with various sugars and base pairing properties. This work is an important advance defining (at least in vitro) nucleotide selection by telomerase through the first high resolution crystal structures of *Triboliumcastaneum* telomerase reverse transcriptase (TERT) throughout its catalytic cycle and characterization of the Tribolium and human TERT kinetics. The authors solved the three structures of Tribolium TERT in complex with a DNA/RNA duplex with 1 nucleotide (nt) overhang, the same duplex with a non-hydrolyzable analog of dGTP and a product DNA/RNA duplex. The most novel finding was a steric gate which allows TERT to discriminate dNTPs from rNTPs and the authors show that this residue (Y717) serves similar functions in human telomerase.

The reviewers and the reviewing editor, however feel that there are insufficient discussion and comparison with other structural and biochemical work, especially in the light of similarity between these new reported structures and those published by Gillis et al., 2008 and Mitchell et al., 2010, with the product state being almost the same as that reported by Mitchell et al., 2010. There is also a question of state I designation that needs to be addressed experimentally.

Essential revisions:

Significant revisions that will require new experiments:

The main concern of the reviewers was related to "state I" designation. The authors chose a substrate where the RNA primer has just one 3' nucleotide overhang, perhaps to prevent subsequent cycles of catalysis. But since *tc*TERT uses RNA primers with up to 6 nt overhang, it is conceivable that the template-TERT conformation, especially the RNA-interacting residues could be very different between 1 and 6 templating bases. In a natural telomerase RNP, the RNA template threads through the TERT ring. The reviewers questioned whether "state I" described in the paper with only 1 nt overhang truly represents state I. In the recent Tetrahymena EM structure at 4.8 Å (Jiang et al., Science 2018), a stretch of the locked-DNA substrate is bound to the RNA template, which may be more representative of state I in the cycle. For the main conclusions of this paper to be valid, the authors need to provide a new structure of state I with the 6 nt overhang RNA primer (preferred) or show that the structure they have behaves the same way as State I.

Other significant revisions:

1) Wu et al., (EMBO 2017) extensively characterized how the DNA substrate is handled during each cycle of repeat synthesis and suggested a more detailed catalytic cycle than what is proposed in Figure 1 here. It involved partial duplex melting before the synthesis reaches the template 5' end, allowing translocation to occur. It has also been shown that the kinetics of the addition of each nt within each telomeric repeat is not the same (Chen et al., 2018). So far the structures and kinetics have only been done with one single addition of dGTP in this work, which will be different for other nucleotides of the repeat. This point needs to be carefully addressed.

2) The authors identified critical residues for the catalytic activity and fidelity, revealing similarities and differences with other DNA polymerases. This said, while Tribolium TERT provides a good model system for studying TERT structure, it lacks the TEN domain and a large part of the IFD, which have been shown to be crucial for TERT activity and processivity in human and Tetrahymena. This major difference and whether the kinetics observations can be generalized to other telomerases warrants a discussion.

3) The Y717 steric gate is required for discriminating dNTPs from rNTPs, which is nicely shown in the human telomerase case. There is indeed a correlation between the number nucleotide in the repeat and the rate of incorporation. However, it is intriguing that the enzyme is still able to incorporate a couple of repeats with rATP. This suggests that in the human case the rATP in this position can be tolerated to some extent. Besides the sugar gate, could the base identity have some effects too?

4) The description of critical residues surrounding RNA as well as incoming nucleotide is often too vague. More modeling or description is needed. For instance, how does R194 relocation/move between different state?

5) Please provide details to those who are not familiar with *T. casteneum* telomerase such as the TERT and TR molecular weights and how they differ from human or mouse telomerase. Also the authors should elaborate more on the TR version for *T. casteneum*. We assume as readers that the authors could not co-crystalize with "tcTR" for similar reasons as the field is not able to crystalize human hTR + hTERT. Please provide some background on that (perhaps in the Introduction).

6) It seems that the authors generated the RNA:DNA hybrid substrate prior to incubating with *tc*TERT. What was the reason for this? Could they incubate the RNA 16-mer with *tc*TERT to allow that complex to assemble and then add the 15-mer complimentary DNA substrate? Would they see the same results? Technically speaking the substrate for telomerase is the DNA (nucleotide or single stranded). The RNA is a part of the enzymatic complex.

7) Subsection “Product complex”. It makes sense that there are minimal rearrangements since it could be predicted that the majority of movement is in the RNA subunit of telomerase (which is missing in this model). However, a key question is to know how it stays on and translocates after a full hexameric repeat is added. The authors may want to review a recent paper (J. Biol. Chem., 294(30):11579-11588, 2019). This work provides a more macro look at the telomerase catalytic cycle that may provide some insights into the present studies.

8) Please elaborate on the chemistry step vs nucleotide binding step.

9) Subsection “The steric gate of telomerase” paragraph three. Please include the details of the telomerase used. Was a recombinant telomerase with 3xFLAG tag and overexpressed hTR used? This is important as we have now seen that the tags used for telomerase play a role for in-vitro and in-vivo activity.

10) Figure 5C. If you generated a custom single-stranded substrate with dNTPs and rNTPs, would you even lose that first extension? Maybe an experiment like this will test the assumption that it "may inhibit the telomerase translocation step."

11) Subsection “The steric gate of telomerase” paragraph four. rGTP displayed the greatest inhibition. Is this because 3Gs are in the hexameric repeat or is this due to something else? Some discussion on this would be helpful.

12) While all alterations resulted in reduction of telomere extension by telomerase, did any alterations tested enhance telomere extension (e.g. decreased pausing, improved processivity etc). This should be mentioned in the Discussion.

13) Any evidence in the current series of experiment that the alterations of inserted ribonucleotides causes DNA damage or lack of shelterin protection. While this has been shown by others in cells using altered nucleotide insertion into telomeres, there are no cell-based studies in the present studies to confirm or progress much of the speculation in the Discussion since all experiments in the present studies use a somewhat artificial in vitro (test tube) approach. For example, do altered ribonucleotides persist in telomeres and is there any evidence that RER removes them? Also, what is the evidence that a single misinsertion every 10kb would prevent shelterin binding or disrupt G-quadruplex stability? Even with these concerns, the studies are important to help future studies confirm and extend these finding.

---

## [Author Response]

Essential revisions:Significant revisions that will require new experiments:The main concern of the reviewers was related to "state I" designation. The authors chose a substrate where the RNA primer has just one 3' nucleotide overhang, perhaps to prevent subsequent cycles of catalysis. But since tcTERT uses RNA primers with up to 6 nt overhang, it is conceivable that the template-TERT conformation, especially the RNA-interacting residues could be very different between 1 and 6 templating bases. In a natural telomerase RNP, the RNA template threads through the TERT ring. The reviewers questioned whether "state I" described in the paper with only 1 nt overhang truly represents state I. In the recent Tetrahymena EM structure at 4.8 Å (Jiang et al., Science 2018), a stretch of the locked-DNA substrate is bound to the RNA template, which may be more representative of state I in the cycle. For the main conclusions of this paper to be valid, the authors need to provide a new structure of state I with the 6 nt overhang RNA primer (preferred) or show that the structure they have behaves the same way as State I.

We would like to first apologize to the reviewers and reviewing editor for the confusion in regards to Figure 1A and specifically our nomenclature of “state I”. This figure was generated with simplicity in mind and clearly this was a confusing error on our behalf. To address this error, we have done the following: (1) remade Figure 1A with distinct nomenclature for each step in the catalytic cycle; (2) made an additional supplemental figure (Figure 1—figure supplement 3) that describes all 18 steps during the telomerase catalytic cycle for one telomeric repeat addition; and (3) clearly indicated which structural state we are describing in the telomerase catalytic cycle for each reported structure. These changes are expanded upon below.

We wish to thank the reviewers for their suggestion of an additional structure, and agree that each overhang length from one to six will result in unique TERT structural states, with a combined 18 structures needed in total. To this end, we spent over a year aggressively screening ~70,000 crystallization conditions. This involved screening multiple TERT protein constructs and over 40 different nucleic acid substrates of various different overhang lengths, hairpins, and substrates with overhangs on both sides. Moreover, we also employed the resources and expertise of the KU Lawrence campus structural biology core to screen additional crystallization conditions and protein constructs to no avail. So far, the only nucleic acid substrate that has successfully yielded structural information was the one featured in this paper. Even after this initial hit was obtained, we spent months optimizing the condition, and had to screen through hundreds of crystals at the synchrotron Xray source to get the 2.5 Å resolution presented here. Therefore, even if we were able to immediately get an additional novel crystal hit, a new TERT structure could take upwards of a year to generate at an appreciable resolution. It is worth mentioning, prior to the lab closing down for COVID-19, we were actively working to obtain additional high-resolution structural snapshots with various substrates for follow-up studies and will continue this once the lab and synchrotron reopen as we feel novel structural snapshots at each step is a pressing question in the telomerase field.

Instead of obtaining a new crystal structure for the revised manuscript, we would like to emphasize that, although our structure has a one nucleotide gap instead of six, we feel it provides significant insight into the telomerase catalytic cycle at each step. This is because the pre-nucleotide bound category is visited 6 times (Figure 1A and Figure 1—figure supplement 3, steps A_1-6_) prior to the telomerase translocation step. In other words, although the templating RNA base shifts between each of the 6 insertion positions, the TERT active site must go through the general cycle of: free nucleotide binding pocket → nucleotide triphosphate bound within the active site → nucleotide insertion (i.e. product formation). In the initial submission, we had attempted to group all 18 structural states based on the components in the TERT active site, but in response to these peer reviews we have now clarified our catalytic cycle to explicitly include all 18 states (see Figure 1—figure supplement 3). Furthermore, we have changed the labeling scheme in the text to now identify our structural snapshots as states A_6,_ B_6_, and C_6_. In all other current structures of TERT (including Mitchell et al., 2010, and Jiang et al., 2018), the TERT active site is positioned at the primer terminus, with the last inserted nucleotide still in the nucleotide binding pocket, which are all members of the final product step of nucleotide addition or the “C” category of the structural states in Figure 1—figure supplement 3. We think this new labeling scheme highlights that we are presenting two new categories of structural states, “A” and “B”. Furthermore, although structural information for category A and B structures are unknown (prenucleotide and ternary), we can compare our C category structure to the lower resolution cryo-EM structure of *Tetrahymena thermophila* telomerase (Jiang et al., 2018). The positions of key active site amino acids from *tt*TERT state C_3_ closely align with our state C_6_ (RMSD = 0.83, see Author response image 1, Figure 1—figure supplement 3 for state designations). This structural agreement between different states and homologs implies that active site contacts are similar throughout all 6 overhang lengths.

**Author response image 1. respfig1:** A comparison of ttTERT active site residues with *tc*TERT. A superposition of the *tc*TERT product structure active site (state C6, yellow) with the active site from the product structure from ttTERT (State C3, green, from Jiang et al., 2018, PDB code: 6D6V). Key active site amino acid residues including catalytic residues (in A), and nucleoside coordinating residues (B) align closely, exhibiting a Root Mean Squared Deviation (RMSD) of 0.83.

Finally, previous kinetic studies provide evidence that the structural states that we have presented in this manuscript are more representative of a typical telomerase catalytic cycle than structures of the first insertion event (which we had mistakenly called state I in the initial submission). In a recent *EMBO* paper (2018, Chen et al.), steady-state kinetics were performed for all six templating positions. In this study, it was shown that the K_M_ values for the incoming nucleotide vary by a maximum of ~30-fold depending on the registry position (i.e. overhang length). These values were 120 μM for the 6-nucleotide overhang, and 4, 31, and 5 μM for four, two, and single nucleotide overhangs, respectively. Of note, out of all six templating positions, the substrate with six templating bases exhibited altered kinetics compared to the other 5 positions. Thus, the structural snapshot we report here (one nucleotide overhang) is likely representative of a typical catalytic cycle (i.e. 15/18 structural states of A_2_ to C_6_, Figure 1—figure supplement 3) as compared to a six-nucleotide overhang that is representative of only 3/18 structural states.

In summary, we sincerely apologize for our error in the description of state I in the initial submission and have corrected this in the revised manuscript. While we would like to obtain a structure with a six nucleotide overhang, we are not confident this will be attainable and will likely require a follow-up study on all eighteen catalytic states. Although this structural information is not available, comparisons to TERT structures with a different overhang length (Author response image 1) and the kinetics of various overhang lengths (performed in Chen et al., 2018) give us confidence that the catalytic cycle presented here represents a typical catalytic cycle for telomerase. Furthermore, the overall trends we describe in terms of fidelity, sugar selectivity, and nucleotide binding affinity are based on observed differences of up to four orders of magnitude and likely would not be altered by the moderate ~30-fold changes in K_M_ seen at different registry positions.

Other significant revisions:1) Wu et al., (EMBO 2017) extensively characterized how the DNA substrate is handled during each cycle of repeat synthesis and suggested a more detailed catalytic cycle than what is proposed in Figure 1 here. It involved partial duplex melting before the synthesis reaches the template 5' end, allowing translocation to occur. It has also been shown that the kinetics of the addition of each nt within each telomeric repeat is not the same (Chen et al., 2018). So far the structures and kinetics have only been done with one single addition of dGTP in this work, which will be different for other nucleotides of the repeat. This point needs to be carefully addressed.

We wish to thank the reviewers for highlighting the complexity of the catalytic cycle, and template position dependence of incoming nucleotides. Further clarification about both phenomena have been added to the manuscript text and also below for reference. Additionally, Figure 1 in the manuscript has been carefully updated to better clarify and address the details of the catalytic cycle.

“This core catalytic cycle repeats six times, until a new telomeric repeat is added (Figure 1A, state C_6_). All 18 telomerase states that are required to add one telomeric repeat are shown in Figure 1— figure supplement 3 for reference. Importantly, as the telomerase approaches the end of its template, the DNA:RNA duplex at the 5’ end begins to melt, enabling telomerase to either (1) translocate and anneal the RNA component to the newly extended telomeric repeat, thus allowing for additional repeat addition; or (2) dissociate from the telomeric DNA.”

“Our experiments were carried out specifically with a single dGTP insertion using a 4 nucleotide overhang RNA template. While telomerase has been shown to exhibit moderate base and position-specific effects, our results indicate that the telomerase catalytic core generally exhibits moderate base selection fidelity, similar to that of X-family polymerases involved in DNA repair (Chen et al., 2018; McCulloch and Kunkel, 2008).”

2) The authors identified critical residues for the catalytic activity and fidelity, revealing similarities and differences with other DNA polymerases. This said, while Tribolium TERT provides a good model system for studying TERT structure, it lacks the TEN domain and a large part of the IFD, which have been shown to be crucial for TERT activity and processivity in human and Tetrahymena. This major difference and whether the kinetics observations can be generalized to other telomerases warrants a discussion.

We carefully chose *Tribolium castaneum* TERT as a model TERT with full knowledge that it lacks both the TEN domain and a large part of the IFD, compared to hTERT. We agree and wish to thank the reviewers for suggesting that we explicitly acknowledge and discuss this important difference amongst TERTs. To address this, we have created a new table further entailing the differences between several different TERTs and added text discussing these crucial differences.

“Although TERTs have highly conserved active sites, there are significant changes in the domain architecture between human and *tc*TERT. These include *tc*TERT lacking the N-terminal (TEN) domain and missing a portion of the insertion in fingers domain (IFD) (Supplementary file 1—table 1B). These domains are essential for the activity of other telomerase homologs, and have been hypothesized to be particularly important for telomerase ratcheting during translocation (Steczkiewicz et al., 2011). Therefore, we kept our *tc*TERT kinetics within a single turnover (i.e. insertion) regime, and, wherever possible, complemented the kinetic results with human telomerase studies to characterize the catalytic cycle of telomerase.”

3) The Y717 steric gate is required for discriminating dNTPs from rNTPs, which is nicely shown in the human telomerase case. There is indeed a correlation between the number nucleotide in the repeat and the rate of incorporation. However, it is intriguing that the enzyme is still able to incorporate a couple of repeats with rATP. This suggests that in the human case the rATP in this position can be tolerated to some extent. Besides the sugar gate, could the base identity have some effects too?

We are grateful to the reviewers for commenting that base-identity could also play a role in ribonucleotide tolerance. We originally thought that the number of insertions per repeat was dominating this effect, but agree that base identity and position in the template could also influence ribonucleotide tolerance by telomerase. This is particularly interesting, as rATP is present at a much higher concentration than other ribonucleotides in the cell (Traut, 1994), so it would make sense that more rATPs would be inserted, and therefore need to be tolerated. These ideas have been included in the text.

“In the most extreme case of three rNTPs present per repeat, extension products are evident well into the second repeat, in contrast to the WT telomerase which had almost no insertion events (Figure 5E). We hypothesize that the rGTP insertion drastically inhibits WT telomerase because the first two insertions in the repeat are templated by rC. Therefore, the first event would need to be a rNTP insertion, followed by another rNTP insertion from a potentially unstable primer terminus. We are unable to decipher if inhibitory effects are due to the number of ribonucleotides per repeat, sequence-dependent inhibition, or a combination of both. In contrast to rGTP, telomerase is able to incorporate multiple repeats when rATP is present, albeit at a reduced efficiency compared to dNTPs (Figure 5E).”

4) The description of critical residues surrounding RNA as well as incoming nucleotide is often too vague. More modeling or description is needed. For instance, how does R194 relocation/move between different state?

Thank you to the reviewers for this comment. We welcome a chance to expand a bit more on our structural results, and have added more descriptions throughout that section, including how R194’s position and environment change throughout the catalytic cycle. Please see changes to the Results section.

5) Please provide details to those who are not familiar with T. casteneum telomerase such as the TERT and TR molecular weights and how they differ from human or mouse telomerase. Also the authors should elaborate more on the TR version for T. casteneum. We assume as readers that the authors could not co-crystalize with "tcTR" for similar reasons as the field is not able to crystalize human hTR + hTERT. Please provide some background on that (perhaps in the Introduction).

We thank the reviewers for the opportunity to further clarify the differences between TERT homologs. As the TR component of telomerase creates flexibility and heterogeneity in the structure, the reviewers are correct that any telomerase holoenzyme with a TR component is a poor substrate for crystallography, which is why we used a truncated version of the TR. To address this (and point 2 above), we’ve created a supplemental table (Supplementary file 1—table 1B) entailing the differences in domain architecture and molecular weight between several TERTs from different species, in addition to altering the introductory text:

“…third, using a truncated version of the *T. castaneum* telomerase RNA component (TR), we can readily obtain sufficient quantities of isolated, active *tc*TERT for characterization of the telomerase catalytic cycle by pre-steady-state kinetics and X-ray crystallography (Gillis et al., 2008; Nguyen et al., 2018). Although TERTs have highly conserved active sites, there are significant changes in the domain architecture between human and *tc*TERT. These include *tc*TERT lacking the Nterminal (TEN) domain and missing a portion of the insertion in fingers domain (IFD) (Supplementary file 1—table 1B). These domains are essential for the activity of other telomerase homologs, and have been hypothesized to be particularly important for telomerase ratcheting during translocation (Steczkiewicz et al., 2011). Therefore, we kept our *tc*TERT kinetics within a single turnover (i.e. insertion) regime, and, wherever possible, complemented the kinetic results with human telomerase studies to characterize the catalytic cycle of telomerase.”

6) It seems that the authors generated the RNA:DNA hybrid substrate prior to incubating with tcTERT. What was the reason for this? Could they incubate the RNA 16-mer with tcTERT to allow that complex to assemble and then add the 15-mer complimentary DNA substrate? Would they see the same results? Technically speaking the substrate for telomerase is the DNA (nucleotide or single stranded). The RNA is a part of the enzymatic complex.

Thank you for the feedback on the generation of our TERT:RNA:DNA complex. We agree that in a biological context, the RNA component of telomerase is assembled as part of the enzymatic complex before it binds to DNA. Hence, for the human telomerase experiments, we assembled the RNA component with telomerase prior to running any of our human telomerase reactions. In our structural studies with *tc*TERT, we preincubated the RNA and DNA prior to the addition of TERT because a homogenous sample is vital for crystal formation. We hypothesize that mixing TERT with the RNA component prior to adding the DNA could result in excess free DNA strands or a TERT:RNA complex that isn’t bound to the DNA, which could prevent crystal formation. Furthermore, when we compare the position of the RNA strand from our structure to that from a recent cryo-EM structure of telomerase holoenzyme (that was made by incubating TERT:RNA with the DNA strand, Jiang et al., 2018) the position of the RNA strand overlays well between the two structures within the active site of the enzyme (Author response image 2). Therefore, we do not predict that changing the order of assembly greatly impacts the structure of the TERT:RNA:DNA complexes that are formed over the course of days in a crystallization solution.

**Author response image 2. respfig2:** RNA used with *tc*TERT structures vs TR from *Tetrahymena thermophila*. A superposition of the RNA component from our product structure (yellow, state C6) and the recent cryo-EM structure of *T. thermophila* (green, state C3, Jiang et al., 2018, PDB code 6d6v). *tc*TERT protein is shown as gray ribbons for reference.

For our pre-steady-state kinetic regime, it is of utmost importance that the enzymatic complex is fully assembled prior to introducing the nucleotide triphosphates. This way, the rate limiting steps for catalysis are only the binding of the nucleotide and the subsequent enzymatic turnover; if we only incubated TERT with the RNA template, and then initiated the reaction by adding the DNA strand and the next incoming nucleotide, it is likely that at that point the rate limiting step would be the annealing of the two nucleic acid strands rather than nucleotide binding and subsequent insertion. Therefore, in these assays, we also needed to preincubate the duplex prior to starting the reaction.

7) Subsection “Product complex”. It makes sense that there are minimal rearrangements since it could be predicted that the majority of movement is in the RNA subunit of telomerase (which is missing in this model). However, a key question is to know how it stays on and translocates after a full hexameric repeat is added. The authors may want to review a recent paper (J. Biol. Chem., 294(30):11579-11588, 2019). This work provides a more macro look at the telomerase catalytic cycle that may provide some insights into the present studies.

Thank you for directing our attention to this recent publication; we agree that the macro look at the catalytic cycle is relevant to the present studies, and have made note of some of the key findings of this paper in our introduction of the catalytic cycle. As we see an inactivation of telomerase upon insertion of ribonucleotides, an outstanding question is whether the mechanism of inactivation is related to the catalysis-dependent inactivation observed in the Sayed et al. study. One possibility is that reactivation with iTAFs would increase translocation efficiency with noncanonical substrates. We do feel that these questions, although interesting, lie outside of the scope of the present manuscript.

“Importantly, as the telomerase approaches the end of its template, the DNA:RNA duplex at the 5’ end begins to melt, enabling telomerase to either (1) translocate and anneal the RNA component to the newly extended telomeric repeat, thus allowing for additional repeat addition; or (2) dissociate from the telomeric DNA. The number of times that a single telomerase enzyme traverses this catalytic cycle is tightly regulated. It was recently shown telomerase becomes inactive after two repeats, but can be reactivated by the recently discovered intracellular telomerase-activating factors (iTAFs) (Sayed et al., 2019).”

8) Please elaborate on the chemistry step vs nucleotide binding step.

We have elaborated on these two steps in context of the catalytic cycle for clarity:

“Next, the binary complex binds an incoming dNTP and samples for proper Watson-Crick base pairing to the RNA template (Figure 1A, state B_1_). The transition between these two states represents the nucleotide binding step, measured as a dissociation constant (K_d_). If the resulting ternary complex (TERT:DNA:dNTP) is in the proper orientation, TERT will catalyze the formation of a phosphodiester bond and extend the telomere by one nucleotide (Figure 1A, state C_1_). The transition between these two states is the chemistry step, and its theoretical maximum rate with saturating nucleotide concentration is described as k_pol_. Following insertion of the incoming nucleotide, telomerase will shift registry to align the active site with the next templating base (forming state A_2_).”

9) Subsection “The steric gate of telomerase” paragraph three. Please include the details of the telomerase used. Was a recombinant telomerase with 3xFLAG tag and overexpressed hTR used? This is important as we have now seen that the tags used for telomerase play a role for in-vitro and in-vivo activity.

Yes, the telomerase used was a recombinant telomerase with 3xFLAG tag and overexpressed hTR (see below). For clarity on the construct used, more details of the methods were added to the Results section:

“In these assays, 1.5 telomeric repeats with the sequence TTAGGGTTAG were incubated with 50 µM of either all four dNTPs or all four rNTPs and purified 3xFLAG tagged human telomerase overexpressed with hTR.”

10) Figure 5C. If you generated a custom single-stranded substrate with dNTPs and rNTPs, would you even lose that first extension? Maybe an experiment like this will test the assumption that it "may inhibit the telomerase translocation step."

How pre-existing ribonucleotides in a telomere strand might affect even the first extension of telomerase is a particularly interesting question. This type of question is something we became interested in during prior studies with modified nucleotides inserted into telomeres (Fouquerel et al., 2016). In the case of 8oxoguanine or 8-oxodGTP, pre-existing 8-oxoG in the telomere strand increased telomerase activity by disrupting G-quadruplex structures that prevented telomerase binding, whereas 8-oxodGTP that was inserted by telomerase inhibited the activity of telomerase. This example highlights the complexity of characterizing the effect of inserted non-canonical nucleotides vs pre-existing modified nucleotides in an annealed oligo.

In response to this question, we refer the reviewers to a publication that examined the impact of ribonucleotides in the template for telomerase (Collins and Greider, 1995, *EMBO*). Briefly, telomerase primer extension assays were performed using primers containing various amounts of ribonucleotides present in the primer strand (see Figure 6 in the referenced work). As predicted by the reviewers, in some cases, placing ribonucleotides in the primer strand inhibited even the first extension of telomerase. In other cases, such as the oligonucleotide that only had three ribonucleotides present (sequence: d(G_3_T_2_G)_2_G_3_r(U_2_G) ), telomerase extended the substrate with a reduced efficiency. We feel that the insights from these hybrid substrates better inform our interpretation of how ribonucleotides influence the mechanism of telomerase. Therefore, we added a reference to the work to better make our case that ribonucleotides inhibit telomerase translocation:

“This reduction was evident even with a single rNTP present in a telomeric repeat. Furthermore, previous studies have found telomeric substrates containing ribonucleotides can prevent or reduce extension of the first repeat depending on the number and position of ribonucleotides present in the DNA template (Collins and Greider, 1995).”

11) Subsection “The steric gate of telomerase” paragraph four. rGTP displayed the greatest inhibition. Is this because 3Gs are in the hexameric repeat or is this due to something else? Some discussion on this would be helpful.

We agree that rGTP presence alone exhibited the greatest inhibition out of any of the single ribonucleotides tested. We hypothesize that the inhibition by rGTP is caused by factors in the telomeric sequence. Because the first templating base is an rC in our experimental setup, the first insertion event will be rGTP. In cases where that is successful, the next base in the sequence is also an rC, requiring the extension from a potentially unstable primer terminus, followed by yet another rGTP insertion. Therefore, telomerase cannot increase telomere length much past the initial primer terminus, potentially limiting the binding and keeping it engaged to the telomere end (with only three nucleotides base pairing). With all this in mind, we do concede that unknown effects specific to guanine could also be occurring, but we cannot test these effects without altering the telomere and TR sequences, which would alter insertion efficiencies of telomerase. We do think this adds to the Discussion, and have expanded on this in the manuscript:

“In the most extreme case of three rNTPs present per repeat, extension products are evident well into the second repeat, in contrast to the WT telomerase which had almost no insertion events (Figure 5E). We hypothesize that the rGTP insertion drastically inhibits WT telomerase because the first two insertions in the repeat are templated by rC. Therefore, the first event would need to be a rNTP insertion, followed by another rNTP insertion from a potentially unstable primer terminus. We are unable to decipher if inhibitory effects are due to the number of ribonucleotides per repeat, sequence-dependent inhibition, or a combination of both. In contrast to rGTP, telomerase is able to incorporate multiple repeats when rATP is present, albeit at a reduced efficiency compared to dNTPs (Figure 5E).”

12) While all alterations resulted in reduction of telomere extension by telomerase, did any alterations tested enhance telomere extension (e.g. decreased pausing, improved processivity etc). This should be mentioned in the Discussion.

None of the alterations in this study enhanced telomerase extension and this is a mutation we are very interested in identifying in future studies. We have made a note emphasizing these results in the manuscript:

“Therefore, in order to prevent this telomeric disruption, the telomerase active site appears to have evolved a high degree of stringency towards noncanonical nucleotides, including both rNTPs and mismatched dNTPs. This stringency was evident by the reduced telomere elongation efficiency with every variant tested; other mutations could also be identified with this system that show increases in telomere elongation efficiency.”

13) Any evidence in the current series of experiment that the alterations of inserted ribonucleotides causes DNA damage or lack of shelterin protection. While this has been shown by others in cells using altered nucleotide insertion into telomeres, there are no cell-based studies in the present studies to confirm or progress much of the speculation in the Discussion since all experiments in the present studies use a somewhat artificial in vitro (test tube) approach. For example, do altered ribonucleotides persist in telomeres and is there any evidence that RER removes them? Also, what is the evidence that a single misinsertion every 10kb would prevent shelterin binding or disrupt G-quadruplex stability? Even with these concerns, the studies are important to help future studies confirm and extend these finding.

We would like to thank the reviewer for bringing up these points, and we agree that both the steady-state levels and the biological impacts of ribonucleotides and mismatches in telomere strands are pressing questions brought up by the current study. As such, we have more explicitly outlined the biological question that the present study brings to light. We are already establishing new techniques in our laboratory, including cell culture and modified telomere restriction fragment analysis to answer some of these questions, but they are as of yet unknown, and we believe outside of the scope of the current manuscript.

“While the downstream consequences of telomeric mismatches have not been studied in a biological context to our knowledge, they likely would disrupt G-quadruplex stability and inhibit shelterin protein binding, as both of these phenomena are dependent on DNA sequence (Figure 6B) (Burge, et al., 2006; de Lange, 2005).”

“…[T]elomerase inserts ~40 rNTPs, which represents selectivity comparable to DNA polymerase β and DNA polymerase δ (Brown and Suo, 2011; Cavanaugh, Beard and Wilson, 2010; McElhinny et al., 2010). However, it is unknown whether ribonucleotides persist in telomeres, their biological consequences, and if they are addressed with ribonucleotide excision repair (RER), similar to other genomic ribonucleotides (Sparks et al., 2012).”